# MILP-StuDio: MILP Instance Generation via Block Structure Decomposition

**Haoyang Liu[1], Jie Wang[1]\*, Wanbo Zhang[1], Zijie Geng[1], Yufei Kuang[1], Xijun Li[2, 3],**
**Yongdong Zhang[1], Bin Li[1], Feng Wu[1]**

[1]MoE Key Laboratory of Brain-inspired Intelligent Perception and Cognition,
University of Science and Technology of China
[2] Shanghai Jiao Tong University
[3] Noah's Ark Lab, Huawei Technologies
`{dgyoung,zhang_wb,ustcgzj,yfkuang}@mail.ustc.edu.cn,`
`lixijun@sjtu.edu.cn,`
`{jiewangx,zhyd73,binli,fengwu}@ustc.edu.cn`

## Abstract

Mixed-integer linear programming (MILP) is one of the most popular mathematical formulations with numerous applications. In practice, improving the performance of MILP solvers often requires a large amount of high-quality data, which can be challenging to collect. Researchers thus turn to generation techniques to generate additional MILP instances. However, existing approaches do not take into account specific block structures—which are closely related to the problem formulations—in the constraint coefficient matrices (CCMs) of MILPs. Consequently, they are prone to generate computationally trivial or infeasible instances due to the disruptions of block structures and thus problem formulations. To address this challenge, we propose a novel MILP generation framework, called Block Structure Decomposition (MILP-StuDio), to generate high-quality instances by preserving the block structures. Specifically, MILP-StuDio begins by identifying the blocks in CCMs and decomposing the instances into block units, which serve as the building blocks of MILP instances. We then design three operators to construct new instances by removing, substituting, and appending block units in the original instances, enabling us to generate instances with flexible sizes. An appealing feature of MILP-StuDio is its strong ability to preserve the feasibility and computational hardness of the generated instances. Experiments on commonly-used benchmarks demonstrate that with instances generated by MILP-StuDio, the learning-based solvers are able to significantly reduce over 10% of the solving time.

## 1 Introduction

Mixed-integer linear programming (MILP) is a fundamental mathematical optimization problem that finds extensive applications in the real world, such as scheduling [1], planning [2], and chip design [3, 4]. In industrial scenarios, the solving efficiency of MILPs is associated with substantial economic value. To speed up the solving process, a great number of high-quality MILP instances are required to develop or test the solvers. Here we give the following two examples. First, both traditional solvers [5, 6] and learning-based solvers [7–10] rely heavily on a lot of MILP instances for hyperparameter tuning or model training. Second, evaluating the robustness of solvers needs a comprehensive MILP benchmark consisting of numerous instances. However, acquiring many instances is often difficult due to high acquisition costs or privacy concerns [11, 12]. As a result, the limited data availability poses great challenges and acts as a bottleneck for solver performance.

---

\*Corresponding author. Email: jiewangx@ustc.edu.cn.

38th Conference on Neural Information Processing Systems (NeurIPS 2024).

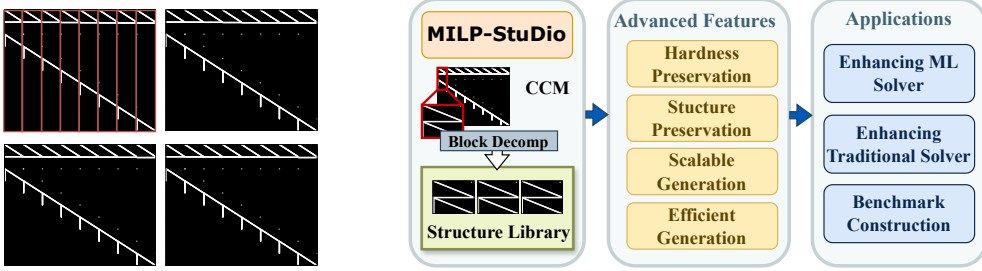

(a) CCMs in FA.

(b) Features and applications of MILP-StuDio.

Figure 1: Figure 1a visualizes the CCMs of four instances from the FA problem, where the white points represent the nonzero entries in CCMs. As we can see, the CCMs exhibit similar block structures across instances, with the patterns in red boxes being the block units. Figure 1b illustrates the block decomposition process, advanced features, and applications of our proposed MILP-StuDio.

This challenge motivates a wide range of MILP generation techniques. In the past, researchers relied on problem-specific techniques for generation [13–17]. These methods assumed knowledge of problem types and generated instances based on the corresponding mathematical formulations, such as the satisfiability problem [18], set covering problem [15], and others. However, these techniques require much expert knowledge to design and are limited to specific MILP problems. They also face limitations in practical scenarios where problem formulations are unknown [19].

In recent years, there has been some progress in general MILP generation that does not require explicit knowledge of the problem formulations. These approaches can be broadly classified into statistics-based and learning-based methods. Statistics-based approaches utilize a few instance statistics to sample in the MILP space [20]. More advanced learning-based approaches, exemplified by G2MILP [19], leverage deep learning models to capture global instance features and iteratively modify constraints in the original instances. Though in the early stage, learning-based techniques offer convenience and strong adaptability for MILP generation, making them applicable in a wider range of practical scenarios [19]. However, they still suffer from significant challenges. (1) They fail to account for the inherent problem structures adequately and disrupt instances' mathematical properties. This leads to low-quality instances with degrading computational hardness or infeasible regions. (2) Existing methods fail to generate instances with different sizes from the original ones, limiting instance diversity. (3) The iterative style to modify constraints becomes time-consuming when dealing with large-scale instances.

Therefore, a natural question arises: can we analyze and exploit the problem structures during generation to address the above challenges? Consider a MILP instance with constraints $Ax \leq b$, where $A$ is the constraint coefficient matrix (CCM), $x$ is the decision variable and $b$ is a vector. As shown in Figure 1a, we observe that a great number of real-world MILP problems exhibit structures with repeated patterns of block units in their CCMs. In operational research, researchers have long noticed the similar block structures of CCMs across instances from the same problem type, and they have been aware of the critical role of CCMs in determining problem formulation and mathematical properties [21–24]. Although a wide suite of CCM-based techniques have been developed to solve MILPs [25–27], existing works on MILP generation rarely pay attention to CCMs. Consequently, these works fail to preserve the block structures during the generation process.

In light of this, we propose a novel MILP generation framework called Block Structure Decomposition (MILP-StuDio), which takes into account the block structures throughout the generation process and addresses Challenge (1)-(3) simultaneously. Specifically, MILP-StuDio consists of three key steps. We begin by identifying the block structures in CCMs and decomposing the instances into block units, which serve as the building blocks in the MILP instances. We then construct a library of the block units, enabling efficient storage, retrieval, and utilization of the comprehensive block characteristics during the subsequent process. Leveraging this library, we design three block operators on the original instances to generate new ones, including block reduction (eliminating certain blocks from the original instances), block mix-up (substituting some blocks with others sampled from the library), and block expansion (appending selected blocks from the library). These operators enable us to generate instances with flexible sizes, effectively improving the diversity of instances.

Experiments demonstrate that MILP-StuDio has the following advanced features. (1) Hardness preservation. MILP-StuDio can effectively preserve the computational hardness and feasibility in the

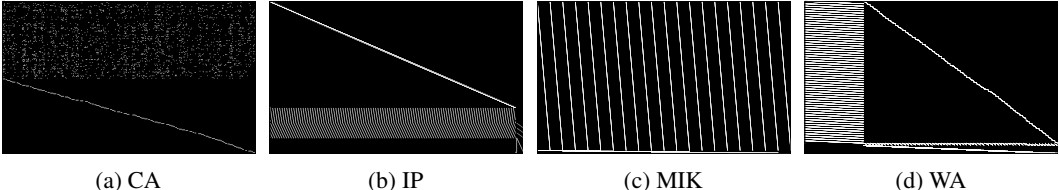

| (a) CA | (b) IP | (c) MIK | (d) WA |

Figure 2: Visualization of the CCMs of instances in four widely recognized benchmarks. The block structures can be commonly seen in MILP problems.

generated instances. (2) Scalable generation. MILP-StuDio can generate instances with flexible sizes. (3) High efficiency. MILP-StuDio can reduce over two-thirds of the generation time in real-world large datasets. We observe an over 10% reduction in the solving time for learning-based solvers using instances generated by MILP-StuDio.

## 2 Background

### 2.1 MILP and MILP with Block Structure

A MILP instance takes the form of:

$$\min_{\boldsymbol{x} \in \mathbb{R}^n} \quad \boldsymbol{c}^\top \boldsymbol{x}, \quad \text{s.t.} \quad \boldsymbol{A}\boldsymbol{x} \leq \boldsymbol{b}, \boldsymbol{l} \leq \boldsymbol{x} \leq \boldsymbol{u}, \boldsymbol{x} \in \mathbb{Z}^p \times \mathbb{R}^{n-p}. \tag{1}$$

In Formula (1), $\boldsymbol{x}$ denotes the decision variables, $\boldsymbol{c} \in \mathbb{R}^n$ denotes the coefficients in the objective function, $\boldsymbol{A} \in \mathbb{R}^{m \times n}$ is the constraint coefficient matrix (CCM) and $\boldsymbol{b} \in \mathbb{R}^m$ denotes the terms on the right side of the constraints, respectively. The vectors $\boldsymbol{l} \in (\mathbb{R} \cup \{-\infty\})^n$ and $\boldsymbol{u} \in (\mathbb{R} \cup \{+\infty\})^n$ denote lower and upper bounds for the variables, respectively.

In real-world applications, a significant portion of MILPs exhibit block structures—consisting of many block units—in their constraint coefficient matrices (CCMs) $\boldsymbol{A}$. These problems, referred as MILPs with block structures, include many commonly-used and widely-studied datasets in recent papers on learning-based solvers [8–10, 28], such as combinatorial auctions (CA), capacitated facility location (FA), item placement (IP), multiple knapsacks (MIK), and workload balancing (WA). In Figure 2, we visualize the CCMs of MILP instances using a black-and-white digital *image representation* [29]. In this representation, the rows and columns of the digital images correspond to the constraints and variables in the MILPs, respectively. To construct the digital image, we assign a pixel value of 255 (white) to the entry $(i, j)$ if the corresponding entry in the CCM $\boldsymbol{A}[i, j]$ is nonzero. Conversely, if $\boldsymbol{A}[i, j]$ is zero, we set the pixel value to 0 (black). This mapping allows us to depict the sparsity patterns and structural characteristics of CCMs visually. For each problem, the CCMs of the instances present a similar block structure, characterizing specific mathematical formulations.

The importance of block-structured CCMs in the context of MILP solving has long been acknowledged by operational researchers, where instances with similar block structures share similar mathematical properties [29–31]. Furthermore, the block structures of CCMs are closely related to the problem formulations [30]. Thus, the block matrices have shown great potential in accelerating the solution process for a family of MILP problems [21–24]. One notable technique developed to exploit this structure is Dantzig-Wolfe decomposition [25] for solving large-scale MILP instances.

### 2.2 Bipartite Graph Representation of MILPs

A MILP instance can be represented as a weighted bipartite graph $\mathcal{G} = (\mathcal{W} \cup \mathcal{V}, \mathcal{E})$ [8]. The two sets of nodes $\mathcal{W} = \{w_1, \cdots, w_m\}$ and $\mathcal{V} = \{v_1, \cdots, v_n\}$ in the bipartite graph correspond to the MILP's constraints and variables, respectively. The edge set $\mathcal{E} = \{e_{ij}\}$ comprises edges, each connecting a constraint node $w_i \in \mathcal{W}$ with a variable node $v_j \in \mathcal{V}$. The presence of an edge $e_{ij}$ is determined by the coefficient matrix, with $\boldsymbol{e}_{ij} = (e_{ij})$ as its edge feature, and an edge $e_{ij}$ does not exist if $\boldsymbol{A}[i, j] = 0$. Please refer to Appendix I.1 for more details on the graph features we use in this paper.

## 3 Motivated Experiments

Preserving the mathematical properties of the original instances is a fundamental concern in MILP generation [19]. These properties encompass feasibility, computational hardness, and problem structures, with the latter being particularly crucial. The problem structure directly determines the problem formulation and, consequently, impacts other mathematical properties. However, it

is important to note that the term "problem structure" can be ambiguous and confusing. There are different understandings and definitions of the problem structure—such as the bipartite graph structure [19] and the CCM's block structure [29]—and we are supposed to identify the most relevant and useful ones that contribute to the mathematical properties of MILPs. Analyzing and exploiting these specific structure types become key factors in improving the quality of the generated instances.

## 3.1 Challenges of Low-Quality Generation

G2MILP is the first learning-based approach for MILP instance generation. While it has shown promising performance, we observe that G2MILP still encounters difficulties in generating high-quality instances for MILPs with block structures. To evaluate its performance, we compare the graph structural distributional similarity and solving properties of the original and generated instances from the workload appointment (WA) benchmark [32] using Gurobi [5], a state-of-the-art traditional MILP solver. We set the masking ratio of G2MILP—which determines the proportion of constraints to be modified—to 0.01. The results are summarized in Table 1. In this table, the *Similarity* metric refers to the graph structural distributional similarity score [19] (defined in Appendix I.3), *Time* represents the average solving time (with a time limit 1,000s), and *Feasible ratio* indicates the proportion of feasible instances out of the total instances. Results show that although the generated instances achieve a high similarity score, most of them are infeasible. Furthermore, the feasible instances exhibit a severe degradation in computational hardness.

Table 1: The comparison of graph similarity and solving properties between the original instances and the generated instances using G2MILP [19], a popular MILP generation framework. Here we use 100 original instances to generate 1,000 instances.

|          | Similarity | Time    | Feasible Ratio |
|----------|------------|---------|----------------|
| Original | 1.000      | 1000.00 | 100.00%        |
| G2MILP   | 0.854      | 12.01   | 10.00%         |

## 3.2 Visualization of CCMs

The aforementioned experiments provide evidence that by iterative modifications of sampled constraints, G2MILP can generate MILP with high graph structural distributional similarities to the original instances, but may lead to disruptions in mathematical properties. Consequently, it becomes necessary to explore alternative definitions of problem structures that offer stronger correlations to the mathematical properties of the instances and can be effectively preserved during the generation process. One such promising structure is the block structures within CCMs.

The concept of block structures within CCMs originates from traditional operational research and has proven to be effective for problem analysis [21–24]. It is widely recognized that MILPs with similar block structures in CCMs often share similar formulations, resulting in similar mathematical properties [29]. We visualize and compare the CCMs of the original and generated instances in Figure 3. In the middle figure, we observe that G2MILP breaks the block pattern in the left and introduces a noisy block in the bottom right. It becomes evident that the generation operation in G2MILP breaks the block structures presenting in the original instances. Thus, it motivates us that exploring and preserving the block structures in CCMs can hold the potential for generating high-quality instances.

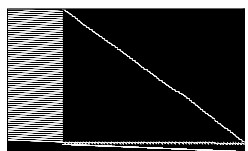 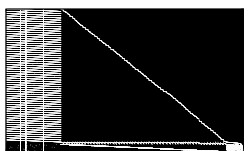 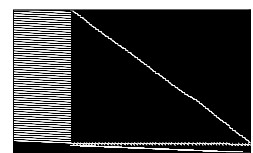

Figure 3: Visualization of CCMs from original instances (left), instances generated by G2MILP (middle), and instances generated by MILP-StuDio (right).

## 4 Generation Framework Using Block Structure Decomposition

In this section, we introduce the proposed MILP-StuDio framework to generate high-quality instances. MILP-StuDio comprises three steps: block decomposition, construction of structure library, and block manipulations. We begin by presenting the concept of block decomposition for MILP in Section 4.1,

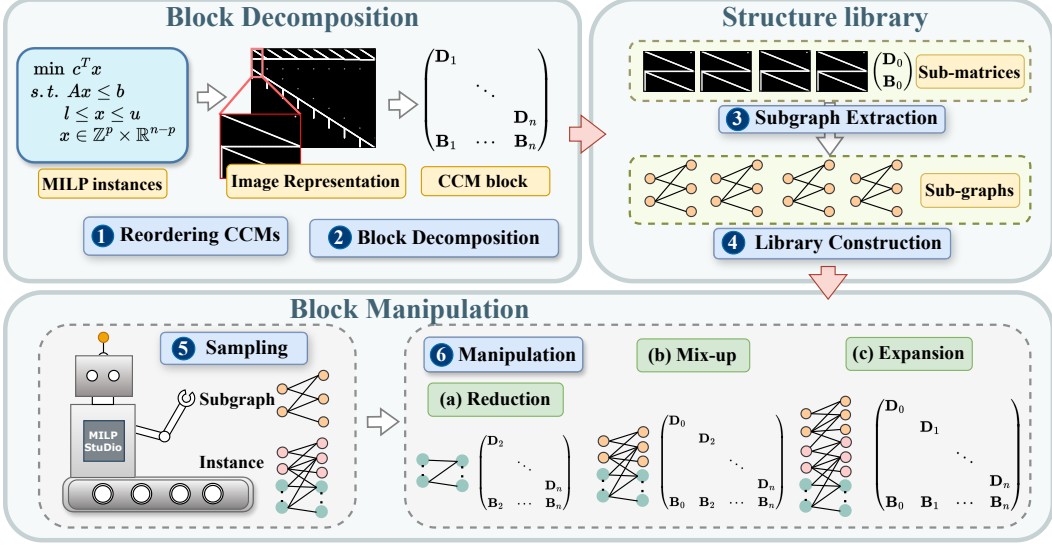

Figure 4: An overview of MILP-StuDio. (1) We detect the block structures in the original instances and decompose the CCMs into sub-matrices of block units. (2) The sub-matrices are transferred into the corresponding sub-graphs of instances' bipartite graph representations. These sub-graphs are used to construct the structure library. (3) We sample instances and sub-graphs of block units and perform block manipulations, including block reduction, mix-up and expansion.

as it forms the core of our method. The subsequent sections, from Section 4.2 to Section 4.4, provide a detailed explanation of each step. The overview of MILP-StuDio is depicted in Figure 4.

## 4.1 Block Decomposition of MILP

In this part, we specify the block structures that we are interested in. CCMs are often reordered to achieve the well-studied block structures [30], including the block-diagonal (BD), bordered block-diagonal (BBD), and doubly bordered block-diagonal (DBBD) structures. We highlight the *block unit* of the decomposition in blue in Equation (2). We can see that the former two structures are special cases of the latter one. Despite the simplicity, they are the building blocks for more complex block structures and are widely used in operational research [30].

$$
\begin{pmatrix} \textcolor{blue}{D_1} & & & \\ & D_2 & & \\ & & \ddots & \\ & & & D_k \end{pmatrix}
\quad
\begin{pmatrix} \textcolor{blue}{D_1} & & & \\ & D_2 & & \\ & & \ddots & \\ & & & D_k \\ \textcolor{blue}{B_1} & B_2 & \cdots & B_k \end{pmatrix}
\quad
\begin{pmatrix} \textcolor{blue}{D_1} & & & & \textcolor{blue}{F_1} \\ & D_2 & & & F_2 \\ & & \ddots & & \vdots \\ & & & D_k & F_k \\ \textcolor{blue}{B_1} & B_2 & \cdots & B_k & C \end{pmatrix}
\quad (2)
$$

$(a)$ Block-diagonal  $\qquad$  $(b)$ Bordered block-diagonal  $\qquad$  $(c)$ Doubly bordered block-diagonal

Formally, a MILP with a DBBD structure can be written as

$$
\begin{aligned}
\min_{\boldsymbol{x} \in \mathbb{R}^n} \quad & \boldsymbol{c}_1^\top \boldsymbol{x}_1 + \boldsymbol{c}_2^\top \boldsymbol{x}_2 + \cdots + \boldsymbol{c}_k^\top \boldsymbol{x}_k + \boldsymbol{c}_{k+1}^\top \boldsymbol{x}_{k+1}, \\
\text{s.t.} \quad & \boldsymbol{D}_i \boldsymbol{x}_i + \boldsymbol{F}_i \boldsymbol{x}_{k+1} \le \boldsymbol{b}_i, \quad 1 \le i \le k, \quad \text{(B-Cons if } \boldsymbol{F}_i = \boldsymbol{O}, \text{ otherwise DB-Cons)} \\
& \sum_{i=1}^k \boldsymbol{B}_i \boldsymbol{x}_i + \boldsymbol{C} \boldsymbol{x}_{k+1} \le \boldsymbol{b}_{k+1}, \quad \text{(M-Cons)} \\
& \boldsymbol{l} \le \boldsymbol{x} \le \boldsymbol{u}, \quad \boldsymbol{x} \in \mathbb{Z}^p \times \mathbb{R}^{n-p},
\end{aligned} \quad (3)
$$

where the partition $\boldsymbol{c} = (\boldsymbol{c}_1^\top, \cdots, \boldsymbol{c}_{k+1}^\top)^\top$, $\boldsymbol{x} = (\boldsymbol{x}_1^\top, \cdots, \boldsymbol{x}_{k+1}^\top)^\top$ and $\boldsymbol{b} = (\boldsymbol{b}_1^\top, \cdots, \boldsymbol{b}_{k+1}^\top)^\top$. To process more complex block structures beyond the three basic ones, we specify different types of constraints and variables in a CCM (and the corresponding MILP). First, we classify variables as *block* and *bordered* variables (Bl-Vars and Bd-Vars). The block variables DBBD are $\boldsymbol{x}_{\text{block}} = (\boldsymbol{x}_1^\top, \cdots, \boldsymbol{x}_k^\top)^\top$, which are involved in the blocks $\boldsymbol{D}_i$, $(1 \le i \le k)$. The bordered variables are

defined to be those in $\boldsymbol{F}_i$, $(1 \leq i \leq k)$, i.e. the variables $\boldsymbol{x}_{k+1}$. Notice that all the variables in BD and BBD are block variables. Then, we classify the constraints in an instance as *master*, *block* and *doubly block constraints* (M-Cons, B-Cons, and DB-Cons), which we have illustrated in Equation (2). As we can see, BD only contains B-Cons, BBD contains B-Cons and M-Cons, and DBBD contains DB-Cons and M-Cons. The classifications of constraints and variables make it possible for us to investigate more delicate structures in the instances—such as the combination of the three basic ones—in the subsequent process (please see Appendix H.2 and H.4).

## 4.2 Block Decomposition

**Reordering Rows and Columns in CCMs**   Given a MILP instance, the raw orders of rows and columns for a CCM are determined by the orders of constraints and variables respectively, which is defined when we establish the instance. Generally, the block structures are not readily apparent in this raw form. To identify and exploit these block structures, we employ a structure detector implemented in the Generic Column Generation (GCG) solver [33] for CCM reordering. This detector identifies row and column permutations that effectively cluster the nonzero coefficients, thereby revealing distinct block structures within CCMs.

**Block Decomposition**   We employ an enhanced variable partition algorithm based on the image representations of CCMs for block decomposition, using the constraint-variable classification results mentioned in Section 4.1. Specifically, we extract the sub-matrices of the *block units* $\mathcal{BU}$ in CCMs, i.e., which take the form of $\boldsymbol{D}_i$ in BD, $\begin{pmatrix} \boldsymbol{D}_i \\ \boldsymbol{B}_i \end{pmatrix}$ in BBD, and $\begin{pmatrix} \boldsymbol{D}_i & \boldsymbol{F}_i \\ \boldsymbol{B}_i & \end{pmatrix}$ in DBBD. Finally, we partition and decompose the CCMs into sub-matrices of block units. The algorithm enables us to handle more complex structures beyond the basic three found by GCG, such as instances in WA with M-Cons, B-Cons and BD-Cons. In the case of WA, the sub-matrices are in the form of $\begin{pmatrix} \boldsymbol{D}_i^{(1)} & \boldsymbol{F}_i \\ \boldsymbol{B}_i & \\ \boldsymbol{D}_i^{(2)} & \end{pmatrix}$,

where $\boldsymbol{D}_i^{(1)}$ represents the diagonal block with DB-Cons, and $\boldsymbol{D}_i^{(2)}$ represents the diagonal block with B-Cons. Please refer to Section H.3 for the detailed implementation of the decomposition process.

## 4.3 Construction of Structure Library

As we can see in Figure 1a, the block units across instances exhibit striking similarities in terms of the internal structures. These common characteristics indicate that the distribution of block units holds valuable information about the problem formulations, making it an ideal building block for reconstructing new instances. Given block-unit sub-matrices of CCMs obtained in Section 4.2, we proceed to extract the corresponding bipartite sub-graphs within the graph representations of the original instances. Compared to the image representation, graph representation offers more convenience for modifying MILP instances during block manipulation. Specifically, suppose that a sub-matrix contains constraints $\tilde{\mathcal{W}} = \{w_{i_1}, \cdots, w_{i_k}\}$ and variables $\tilde{\mathcal{V}} = \{v_{i_1}, \cdots, v_{i_l}\}$ in the instances, we then extract the sub-graph containing $\tilde{\mathcal{W}}$, $\tilde{\mathcal{V}}$ and the edges connecting $\tilde{\mathcal{W}}$ and $\tilde{\mathcal{V}}$ in the bipartite graph representation of the original instance. Subsequently, we collect these sub-graphs from all the training instances and utilize them to construct a comprehensive structure library denoted as $\mathcal{L}$. This structure library serves as a repository for the collected sub-graphs, allowing efficient storage, retrieval, and utilization of the block information.

## 4.4 Scalable Generation via Block Manipulations

With the structure library, we devise three types of generation operators that enable the generation of high-quality MILP instances with flexible sizes. These operators, namely block reduction, block mix-up, and block expansion, play a crucial role in the instance generation process.

- **Reduction.**   This operator involves randomly sampling a block unit $\mathcal{BU}_{\text{ins}}$ from the original instances and then removing it. The reduction operator generates MILP instances with smaller sizes compared to the original ones, reducing the complexity of the problem.
- **Mix-up.**   This operator involves randomly sampling one block unit $\mathcal{BU}_{\text{ins}}$ from the original instances and another block unit $\mathcal{BU}$ from the structure library $\mathcal{L}$. We then replace $\mathcal{BU}_{\text{ins}}$ with $\mathcal{BU}$ to generate a new instance. The mix-up operator introduces structural variations through the incorporation of external block units.

- **Expansion.** This operator involves randomly sampling a block unit $\mathcal{BU}$ from the structure library $\mathcal{L}$ and appending it to the original instances. This process generates new instances of larger sizes compared to the original ones, potentially introducing more complex structures.

To preserve the block structures, the operators should leverage the constraint-variable classification results. Taking the expansion operator as an example, the coefficients of M-Cons in the external block unit should be properly inserted into the M-Cons of the original instances. Meanwhile, we construct new constraints for the B- and DB-Cons of the block using the corresponding right-hand-side terms $b_i$. Finally, we design a coefficient refinement algorithm to align the coefficients of the external blocks during mix-up and expansion (please see Appendix H.4 for details).

## 5 Experiments

### 5.1 Experiment Settings

**Benchmarks**  We consider four MILP problem benchmarks: combinatorial auctions (CA) [16], capacitated facility location (FA) [17], item placement (IP) [32] and workload appointment (WA) [32]. The first two benchmarks, CA and FA, are commonly-used benchmarks proposed in [8]. The last two benchmarks, IP and WA, come from two challenging real-world problem families used in NeurIPS ML4CO 2021 competition [32]. The numbers of training, validation, and testing instances are 100, 20, and 50. More details on the benchmarks are in Appendix I.2.

**Metrics**  We leverage three metrics to evaluate the similarity between the original and generated instances. (1) Graph statistics are composed of 11 classical statistics of the bipartite graph [34]. Following [19], we compute the Jensen-Shannon divergence for each statistic between the generated and original instances. We then standardize the metrics into similarity scores ranging from 0 to 1. (2) Computational hardness is measured by the average solving time of the instances using the Gurobi solver [5]. (3) Feasible ratio is the proportion of feasible instances out of the total ones.

**Baselines**  We consider two baselines for MILP generation. The first baseline is the statistics-based MILP generation approach Bowly [20], which generates MILP instances by controlling specific statistical features, including the coefficient density and coefficient mean. We set these features to match the corresponding statistics of the original instances so as to generate instances with high statistical similarity with the original ones. The second baseline is the learning-based approach G2MILP [19]. By leveraging masked VAE, G2MILP iteratively masks one constraint node in the bipartite graph and replaces it with a generated one.

**Downstream Tasks**  We consider three downstream tasks to demonstrate the effectiveness of the generated instances in practical applications. (1) Improving the performance of learning-based solvers, including predict-and-search (PS) [10] in Section 5.3 and the GNN approach for learning-to-branch [8] in Appendix F.1. (2) Hyperparameter tuning for the traditional solver (please see Appendix F.3). In the above two tasks, we use MILP-StuDio and the baselines to generate new instances to enrich the training data. We also consider the task of (3) hard instances generation in Section 5.4, which reflects the ability to construct hard benchmarks.

**Variants of MILP-StuDio**  During the generation process, we use the three operators to generate one-third of the instances, respectively. We can choose different modification ratios, which implies that we will remove (substitute or append) block units when performing the reduction (mix-up or expansion) operator until the proportions of variables modified in the instance reaches $\eta$.

### 5.2 Similarity between the Generated and the Original Instances

For each generation technique, we use 100 original instances to generate 1,000 instances and evaluate the similarity between them. As shown in Table 2, we present the graph structural distributional similarity scores between the original and generated instances. We do not consider Bowly in WA since the instance sizes in WA are so large that the generation time is over 200 hours. The results suggest that MILP-StuDio shows a high graph structural similarity compared to the baselines. In FA, instances generated from G2MILP present a low similarity, while our proposed MILP-StuDio can still achieve a high similarity score. As the modification ratio increases, the similarity scores of G2MILP and MILP-StuDio decrease, whereas G2MILP suffers from a more severe degradation.

We also evaluate the computational hardness and feasibility of the generated instances in Table 3. We report the average solving time and feasibility ratio for each dataset. Results demonstrate that the instances generated by the baselines represent a severe degradation in computational hardness.

Table 2: Structural Distributional similarity scores between the generated instances and the original ones. The higher score implies higher similarity. *Timeout* implies the generation time is over 200h.

| | Bowly | $\eta = 0.01$ | | $\eta = 0.05$ | | $\eta = 0.10$ | |
|---|---|---|---|---|---|---|---|
| | | G2MILP | MILP-StuDio | G2MILP | MILP-StuDio | G2MILP | MILP-StuDio |
| CA | 0.567 | 0.997 | 0.997 | 0.995 | 0.981 | 0.990 | 0.946 |
| FA | 0.077 | 0.358 | 0.663 | 0.092 | 0.646 | 0.091 | 0.618 |
| IP | 0.484 | 0.717 | 0.661 | 0.352 | 0.528 | 0.336 | 0.493 |
| WA | Timeout | 0.854 | 0.980 | 0.484 | 0.853 | 0.249 | 0.783 |

Table 3: Average solving time (s) and feasible ratio (in parentheses) of the instances. We set the solving time limit to be 1,000s. We mark the **values closest to those in original instances** in bold.

| | CA | FA | IP | WA |
|---|---|---|---|---|
| Original Instances | 0.50 (100.0%) | 4.78 (100.0%) | 1000 (100.0%) | 1000 (100.0%) |
| Bowly | 0.02 (100.0%) | 0.07 (100.0%) | 56.45 (100.0%) | Timeout |
| G2MILP $\eta = 0.01$ | 0.58 (100.0%) | 0.02 (1.7%) | 802.3 (100.0%) | 12.01 (10.0%) |
| MILP-StuDio $\eta = 0.01$ | **0.50 (100.0%)** | **4.94 (100.0%)** | **1000 (100.0%)** | **1000 (100.0%)** |
| G2MILP $\eta = 0.05$ | 0.69 (100.0%) | 0.01 (1.8%) | 0.14 (100.0%) | 0.01 (0.0%) |
| MILP-StuDio $\eta = 0.05$ | **0.48 (100.0%)** | **4.92 (100.0%)** | **691.66 (100.0%)** | **1000 (100.0%)** |
| G2MILP $\eta = 0.10$ | 0.72 (100.0%) | 0.01 (2.0%) | 0.03 (100.0%) | 0.02 (0.0%) |
| MILP-StuDio $\eta = 0.10$ | **0.40 (100.0%)** | **4.64 (100.0%)** | **550.33 (100.0%)** | **1000 (94.5%)** |

Moreover, most of the instances generated by G2MILP in FA and WA datasets are infeasible. Encouragingly, MILP-StuDio is able to preserve the computational hardness and feasibility of the original instances, making it a strong method for maintaining instances' mathematical properties.

We visualize the CCMs of the original and generated instances to demonstrate the effectiveness of MILP-StuDio in preserving block structures in Appendix G. MILP-StuDio is able to preserve the block structures while all the baselines fail to do so.

### 5.3 Improving the Performance of Learning-based Solvers

GNN branching policy [8] and predict-and-search [10] are two representative approaches of learning-based solvers. Here we report the results of PS built on Gurobi and leave the other in Appendix F.1. In alignment with [10], we consider two additional baselines Gurobi and BKS. For each instance, we run Gurobi on a single-thread mode for 1,000 seconds. We also run Gurobi [5] for 3,600 seconds and denote the obtained objective values as the best-known solution (BKS). For more details on the implementation of PS, please refer to Appendix H.1. Table 4 demonstrates the performance of MILP-StuDio-enhanced PS and the baselines, where we set the modification ratio $\eta = 0.05$. We report three metrics to measure the solving performance. (1) Obj represents the objective values achieved by different methods. (2) $\text{gap}_{abs}$ is the absolute primal gap defined as $\text{gap}_{abs} := |\text{Obj} - \text{BKS}|$, where smaller gaps indicate superior primal solutions and, consequently, a better performance. (3) Time denotes the average time used to find the solutions.

In CA and FA, all the approaches can solve the instances to the optimal with $\text{gap}_{abs} = 0$, thus we compare the Time metric. In IP and WA, all the approaches reach the time limit of 1,000s, thus we mainly focus on the $\text{gap}_{abs}$ metric. Methods built on PS does not perform well in CA compared to Gurobi, since CA is easy for Gurobi to solve and PS needs to spend time for network inference. Even so, PS+MILP-StuDio achieves a comparable performance to Gurobi. In the other three datasets, MILP-StuDio-enhanced PS outperforms other baselines, achieving the best solving time or $\text{gap}_{abs}$.

### 5.4 Hard Benchmark Generation

The hard instance generation task is important as it provides valuable resources for evaluating solvers and thus potentially motivates more efficient algorithms. The objective of this experiment is to test the ability to generate harder instances within a given number of iterations. We use 30 original instances to construct the pool. In each iteration, for each instance in the pool, we employ mix-up or expansion operators to generate two new ones, and We then select the instance among and with the longest solving time to replace in the pool. This setting is to preserve the diversity of the pool. We observe that there exist slight differences in the hardness of the original instances, and the generated instances

Table 4: Comparison of solving performance in PS between our approach and baseline methods, under a $1,000$s time limit. In IP and WA, all the approaches reach the time limit, thus we do not consider the Time metric. The notation *Infeasible* implies that a majority of the generated instances are infeasible and thus cannot be used as the training data for PS. '↑' indicates that higher is better, and '↓' indicates that lower is better. We mark the **best values** in bold.

| | CA (BKS 7453.42) | | | FA (BKS 17865.38) | | | IP (BKS 11.16) | | WA (BKS 698.80) | |
|---|---|---|---|---|---|---|---|---|---|---|
| | Obj ↑ | $\text{gap}_{abs}$ ↓ | Time ↓ | Obj ↓ | $\text{gap}_{abs}$ ↓ | Time ↓ | Obj ↓ | $\text{gap}_{abs}$ ↓ | Obj ↓ | $\text{gap}_{abs}$ ↓ |
| Gurobi | **7453.42** | **0.00** | **0.77** | 17865.38 | 0.00 | 7.36 | 11.43 | 0.27 | 698.85 | 0.05 |
| PS | 7453.42 | 0.00 | 0.94 | 17865.38 | 0.00 | 7.19 | 11.40 | 0.24 | 699.05 | 0.25 |
| PS+Bowly | 7453.42 | 0.00 | 0.93 | 17865.38 | 0.00 | 7.25 | 11.63 | 0.47 | Timeout | |
| PS+G2MILP | 7453.42 | 0.00 | 0.89 | Infeasible | | | 11.55 | 0.39 | Infeasible | |
| PS+MILP-StuDio | 7453.42 | 0.00 | 0.78 | **17865.38** | **0.00** | **7.07** | **11.29** | **0.13** | **698.83** | **0.03** |

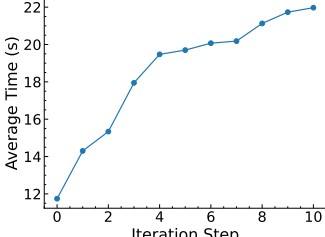

Figure 5: Mean solving time during iterations.

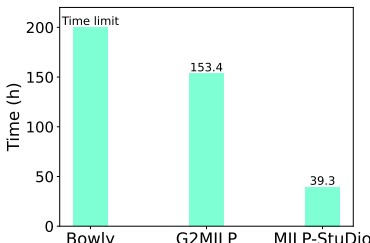

Figure 6: Generation time of 1,000 instances in WA.

derived from the harder original instances are also harder than those from easier ones. If we had simply generated 60 instances and selected the hardest 30, the proportion of instances generated from the hard original instances would have continuously increased, reducing the diversity of the pool. In Figure 5, we depict the growth curve of the average solving time of instances in the pool during 10 iterations. The solving time of the final set is two times larger than that of the initial set, suggesting MILP-StuDio's ability to generate increasingly harder instances during iterations.

### 5.5  Extensive Studies

**Generation Efficiency**   WA is a challenging benchmark with large instance sizes. The instances in WA have over 60,000 constraints and variables, making generation in WA especially time-consuming. We compare the generation time of the generation techniques to generate 1,000 instances. We set the time limit to 200 hours. The results in Table 6 show that MILP-StuDio significantly archives $3\times$ acceleration compared to G2MILP, demonstrating high generation efficiency.

**More Results**   We conduct ablation studies on three operators and modification ratios in Appendix F.2, and we also try to extend MILP-StuDio to MILPs without block structures in Appendix F.4 and MILPs in the real-world industrial dataset in Appendix F.5.

## 6  Related Work on MILP Generation

The field of MILP generation encompasses two main categories: problem-specific generation and general MILP generation. Problem-specific generation methods rely on expert knowledge of the mathematical formulation of problems to generate instances. Examples include set covering [15], combinatorial auctions [16], and satisfiability [18]. While these methods can generate instances tailored to specific problem types, they are limited in their applicability and require much modeling expert knowledge. On the other hand, general MILP generation techniques aim to generate MILPs using statistical information [20] or by leveraging neural networks to capture instance distributions [19]. G2MILP [19] is the first learning-based generation framework designed for generating general MILPs. This approach represents MILPs as bipartite graphs and utilizes a masked variational auto-encoder [35] to iteratively corrupt and replace parts of the original graphs to generate new ones.

# 7    Limitations and Future Avenue

Our method originates from the field of operational research and is designed for instances with block structures. The performance of the detector in GCG influences the overall generation quality. Although the detector can identify a wide range of useful structures in the real world, it is still limited when facing instances with extremely complex structures. The exploration of enhancing the performance of the detector, such as those involving prior knowledge of the instances, represents a promising avenue for future research.

# 8    Conclusion

In this paper, we propose a novel MILP generation framework (MILP-StuDio) to generate high-quality MILP instances. Inspired by the studies of CCMs in operational research, MILP-StuDio manipulates the block structures in CCMs to preserve the mathematical properties—including the computational hardness and feasibility—of the instances. Furthermore, MILP-StuDio has a strong ability of scalable generation and high generation efficiency. Experiments demonstrate the effectiveness of MILP-StuDio in improving the performance of learning-based solvers.

# 9    Acknowledgement

The authors would like to thank all the anonymous reviewers for their insightful comments and valuable suggestions. This work was supported by the National Key R&D Program of China under contract 2022ZD0119801 and the National Nature Science Foundations of China grants U23A20388 and 62021001.

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

## Table of Contents for Appendix

# A   More Related Work

## A.1   Machine Learning for Solving MILP

In recent years, there has been notable progress in utilizing machine learning approaches to accelerate MILP solvers [36]. The learning-based approaches for solving MILPs can be broadly categorized into two main groups. The first group of works replaces specific components of the solver that greatly impact solving efficiency, such as branching [8, 37–39], cut selection [40, 9, 41–43], node selection [7, 44] and presolve [45, 46]. These approaches are integrated into exact MILP solvers, ensuring that the resulting solutions are optimal. However, they also inherit the limitation of exact solvers, which can be time-consuming for solving large-scale instances.

The second category includes learning-aided search approaches, which encompass techniques such as predict-and-optimize [10, 47, 48], large neighbor search [49, 50] and so on. Among these methods, predict-and-search (PS) [10] has gained significant popularity. In PS, a machine learning model is first employed to predict a feasible solution, which is then used as an initial point for a traditional solver to explore the solution space and find the best possible solution within a defined neighborhood. By leveraging the predicted feasible solution, PS effectively reduces the search space of the solver, leading to accelerated search and the discovery of high-quality solutions.

By leveraging machine learning techniques, these approaches have demonstrated substantial improvements in the solving efficiency and solution quality of MILP problems. However, training such learning-based models requires many MILP instances as training data, which can be challenging to obtain in real-world applications [19]. There is still ongoing research in this area to improve the sample efficiency of these solvers [10] and study the MILP generation techniques [19].

## A.2   MILP with Block Structure

Operational researchers have observed that a great number of real-world MILP problems exhibit block structures within their constraint coefficient matrices (CCMs) [30, 27]. These problems are prevalent in various applications, including scheduling, planning, and knapsack scenarios [31, 16, 17]. Thus, many researchers have focused on understanding the impact of these block structures on the mathematical properties of the instances, and how to leverage them to accelerate the solving process[21–24, 26, 51]. Among this stream of research, the Dantzig-Wolfe decomposition (DWD) [52] stands out as one of the most successful applications. Many classical textbooks on linear programming dedicate sections to discussing this decomposition algorithm [53, 21], underscoring its importance. DWD is widely utilized when a CCM contains both block-diagonals and coupling constraints [52], and can significantly accelerate the solving process for large-scale linear programming and mixed-integer linear programming.

To identify the block structures in a raw CCM, the state-of-the-art detecting algorithm is implemented in GCG (Generic Column Generation) [33], distributed with the SCIP framework [6]. This sophisticated algorithm utilizes row and column permutations to effectively cluster the nonzero coefficients in CCMs, thereby revealing distinct block structures, including the useful block-diagonal, bordered block-diagonal, and doubly bordered block-diagonal structures.

# B   Broader Impacts

This paper introduces a MILP instance generation framework (MILP-StuDio) that aims to generate additional MILP instances with highly similar computational properties to the original instances. Our approach holds significant potential in numerous practical applications and important engineering scenarios, including scheduling, planning, facility location, etc. With instances generated by MILP-StuDio, we can improve the performance of block-box learning-based or traditional solvers at a low data acquisition cost. One notable advantage of our proposed method is that it does not rely on the knowledge of MILP formulations and can be applied to general MILP instances, eliminating concerns regarding privacy disclosure. Furthermore, as a plug-and-play module, MILP-StuDio offers great convenience for users.

## C  The Importance of MILPs with Block Structures

The MILPs with block structures are important in industrial and academic fields. We found that MILP instances with block structures are commonly encountered in practical scenarios and have been an important topic in operations research (OR) with much effort [25, 54–58].

MILP with block structures is an important topic in OR. Analyzing block structures is a critical tool for analyzing the mathematical properties of instances or accelerating the solving process (e.g., Dantzig-Wolfe decomposition [25]) in OR. The MIPLIB dataset also provides visualization results of the constraint coefficient matrices for each instance, highlighting the prevalence of block structures.

The MILP instances with block structures are common and have wide applications in daily production and life. There are many examples where the instances present block structures, including the allocation and scheduling problems [54], the multi-knapsack problem [55], the security-constrained unit commitment problem in electric power systems [56], multicommodity network flow [57], multicommodity transportation problem [58] and so on. In real-world optimization scenarios, there are different types of similar items—such as different workers or machines in planning and scheduling problems, a set of power generation units in the electric power systems, vehicles in the routing problems, and so on—with relevant variables naturally presents a block-structured form in the mathematical models.

The datasets we used in this paper (IP and WA) are from real-world applications. The NeruIPS 2021 Competition of Machine Learning for Combinatorial Optimization [32] released three well-recognized challenging datasets from real-world applications (IP, WA, and the anonymous dataset). Two of the three competition datasets (IP and WA) have block structures. Moreover, instances from the anonymous dataset are selected from MIPLIB with large parts having block structures. These further reflect the wide application of block structures in real-world applications. Thus, our method works in a wide range of problems in practice.

Researchers have investigated specific MILP problems with block structures. MILP with block structures has a large scope in the optimization field and there has been a wide range of works on specific problems with block structures, and they have developed a suite of optimization problems tailored to these problems. For example, the tailored algorithm for the security-constrained unit commitment problem in electric power systems [4], multicommodity transportation problem [6], vehicle routing problem [7], and so on.

Thus, MILP with block structures has a large scope in production and optimization. It has drawn much attention in the industry and academic fields.

## D  Explanation: Why do the MILPs Exhibit Block Structures?

The key reasons why MILP instances exhibit block structures can be summarized as follows.

- Repeated items or entities with similar attributes. In many real-world applications involving scheduling, planning, and packing problems, we often encounter multiple items or entities that share the same type or attributes. For instance, in a scheduling problem, there may be multiple destinations or vehicles that exhibit similar characteristics. Similarly, in a knapsack problem, there can be multiple packages or items that are interchangeable from the perspective of operational research or mathematical modeling.

- Symmetric interactions between different types of items. These repeated items or entities, as well as their interactions, lead to symmetries in the mathematical formulation of the MILP instances. For example, in a scheduling problem, all the vehicles may be able to pick up items from the same set of places and satisfy the demand of the same set of locations.

These symmetries in the problem structure are reflected in the blocks of CCMs, where each block may represent the information associated with a certain vehicle, destination, or other item type.

It is important to note that although MILP-StuDio is designed primarily for MILP instances with block structures, it has a broader application to a wider category of problems, including those with more complex block structures (Appendix H.4), blocks of different sizes (Appendix H.4), and even non-structural MILPs (Appendix F.4).

# E  Introductions the Underlying Learning-Based Solvers

## E.1  Learning to Branch

Branching is a critical component of branch-and-bound (B&B) solvers for mixed-integer linear programming (MILP) problems. It involves selecting a fractional variable to partition the feasible region in each iteration. The effectiveness of the chosen variable and the time required to make the branching decision are crucial factors that heavily impact the size of the branch-and-bound search tree and, consequently, the solver's efficiency. Thus, there have been many efforts to develop effective and efficient branching policies. Among the conventional branching policies, the strong branching policy has been demonstrated to yield the smallest branch-and-bound trees. This policy identifies the fractional variable that provides the largest improvement in the bound before performing the branching operation. However, the evaluation process associated with strong branching involves solving a considerable number of linear relaxations of the original MILP, resulting in unacceptable time and computational costs.

To address the limitations of strong branching, [8] proposes a GNN-based approach that employs behavior cloning to imitate the strong branching policy. In this approach, the B&B process is formulated as a Markov decision process, with the solver acting as the environment. At each step, the branching policy receives the current state $\mathbf{s}$, which includes information on past branching decisions and current solving statistics. Subsequently, the policy selects an action $\mathbf{a}$ from the set of integer variables with fractional values in the current state. The researchers parameterize the branching policy as a GNN model, which serves as a fast approximation of the strong branching policy.

During the training process, we first run the strong branching expert on the training and testing instances to collect the state-action pair $(\mathbf{s}, \mathbf{a})$, forming the training dataset $\mathcal{D}_{\text{train}}$ and testing dataset $\mathcal{D}_{\text{test}}$. The GNN branching policy $\pi_\theta$ is then trained by minimizing the cross-entropy loss

$$L(\theta) = -\frac{1}{|\mathcal{D}_{\text{train}}|} \sum_{(\mathbf{s}, \mathbf{a}) \in \mathcal{D}_{\text{train}}} \log \pi_\theta(\mathbf{a} \mid \mathbf{s}).$$

The imitation accuracy refers to the consistency of the learned GNN policy compared to the strong branching expert on the testing dataset $\mathcal{D}_{\text{test}}$:

$$\text{ACC} = \frac{1}{|\mathcal{D}_{\text{test}}|} \sum_{(\mathbf{s}, \mathbf{a}) \in \mathcal{D}_{\text{test}}} \mathbb{I}\left( \arg\max_{\hat{\mathbf{a}}} \pi_\theta(\hat{\mathbf{a}} \mid \mathbf{s}) = \mathbf{a} \right),$$

where $\mathbb{I}$ is the indicator function defined as $\mathbb{I}(\mathbf{c}) = 1$ if the condition $\mathbf{c}$ is true, and $\mathbb{I}(\mathbf{c}) = 0$ otherwise. The imitation accuracy of the GNN model reflects the similarity between the learned branching policy and the expert policy, and it significantly impacts the solver's efficiency.

## E.2  Predict-and-Search

Different from learning-to-branch, which replaces certain heuristics in a traditional B&B solver, Predict-and-Search (PS) [10] belongs to another category of learning-based solvers that directly predict a feasible solution and subsequently perform the neighborhood search. PS aims to leverage a learning-based model to approximate the solution distribution $p(\boldsymbol{x} \mid \mathcal{I})$ given an instance $\mathcal{I}$ from the instance dataset $\mathcal{D}$. Given an instance $\mathcal{I}$, the solution distribution $p(\boldsymbol{x} \mid \mathcal{I})$ is defined as follows:

$$p(\boldsymbol{x} \mid \mathcal{I}) = \frac{\exp(-E(\boldsymbol{x}, \mathcal{I}))}{\sum_{\tilde{\boldsymbol{x}}} \exp(-E(\tilde{\boldsymbol{x}}, \mathcal{I}))}, \text{ where the energy function } E(\boldsymbol{x}, \mathcal{I}) = \begin{cases} \boldsymbol{c}^\top \boldsymbol{x}, & \text{if } \boldsymbol{x} \text{ is feasible,} \\ +\infty, & \text{otherwise.} \end{cases}$$

In the prediction step, PS employs a GNN model to approximate the solution distribution for the binary variables in a MILP instance. To train the GNN predictor $p_\theta(\boldsymbol{x} \mid \mathcal{I})$, PS adopts the assumption, as described in [39], that the variables are independent of each other, i.e., $p_\theta(\boldsymbol{x} \mid \mathcal{I}) = \prod_{i=1}^{n} p_\theta(x_i \mid \mathcal{I})$. To calculate the prediction target, PS collects a set of $m$ feasible solutions for each instance $\mathcal{I}$, from which a vector $\mathbf{p} = (p_1, p_2, \ldots, p_n)^\top$ is constructed. Here, $p_i = p(x_i = 1 | \mathcal{I})$ represents the probability of variable $x_i$ being assigned the value 1, given the instance $\mathcal{I}$. The GNN predictor then outputs the predicted probability $p_\theta(x_i = 1 | \mathcal{I})$ for each variable. Finally, the predictor $p_\theta$ is trained by minimizing the cross-entropy loss

$$L(\theta) = -\frac{1}{N} \sum_{j=1}^{N} \sum_{i=1}^{n} \left( p_i \log p_\theta(x_i = 1 \mid \mathcal{I}_j) + (1 - p_i) \log(1 - p_\theta(x_i = 1 \mid \mathcal{I}_j)) \right),$$

where $N$ represents the number of instances in the training set.

In the search step, PS performs the neighborhood search based on the predicted partial solution $\hat{x}$. This involves employing a traditional solver, such as SCIP [6] or Gurobi [5], to explore the neighborhood $\mathcal{B}(\hat{x}, \triangle)$ of $\hat{x}$ in search of an optimal feasible solution. Here $\triangle$ represents the trust region radius, and $\mathcal{B}(\hat{x}, \triangle) = \{x \in \mathbb{R}^n \mid \|\hat{x} - x\|_1 \leq \triangle\}$ is the trust region. The neighborhood search process is formulated as the following MILP problem,

$$\min_{x \in \mathbb{R}^n} \quad c^\top x, \quad \text{s.t.} \quad Ax \leq b, l \leq x \leq u, x \in \mathcal{B}(\hat{x}, \triangle), x \in \mathbb{Z}^p \times \mathbb{R}^{n-p}.$$

It is worth noting that the accuracy of the GNN predictor strongly influences the overall solving performance. A reliable and accurate prediction of a feasible solution can effectively reduce the search time. Conversely, an inaccurate prediction may result in an inferior search neighborhood and sub-optimal solution.

## F  Extensive Experiment Results

### F.1  Experiments on Learning to Branch

In this section, we conduct experiments on the learning-to-branch task to further demonstrate the effectiveness of MILP-StuDio in enhancing the learning-based solvers.

**Experiment Setup**  We conduct the learning-to-branch experiments following the setting described in the original paper [8]. In our experiments, we evaluate the performance of the solvers using two benchmark problems: the capacitated facility location (FA) and item placement (IP) problems. The capacitated facility location problem (FA) is chosen as a representative problem that solvers can successfully solve within the given time limit. This problem serves as a benchmark to test the solving efficiency of solvers. The item placement problem (IP) is selected as a benchmark where the solvers struggle to find the optimal solution within the given time limit. By including this challenging problem, we can gain insights into the solvers' ability to find a high-quality primal solution.

The training, validation, and testing instance sets used in this work are identical to those described in the main paper. Specifically, for each benchmark, we use 100 training instances, 20 validation instances, and 50 testing instances. To generate training samples, we execute the strong branching expert on instances in each benchmark, collecting 1,000 training samples, 200 validation samples, and 200 testing samples. For the final evaluation, we test the solving performance on 50 instances.

We implement the model with the code available at `https://github.com/ds4dm/learn2branch`. This code leverages the state-of-the-art open-source solver SCIP 8.0.3 [6] as the backend solver. During testing, the solving time limit for each instance is set to 1,000 seconds. Consistent with the setting in [8], we disabled solver restarts and only allowed the cutting plane generation module to be employed at the root node.

**Baselines**  The first baseline we consider is the GNN model [8], which is trained on an initial set of 1,000 samples collected from 100 original instances (*GNN*). To explore the effectiveness of different instance generation techniques, we use the same set of 100 training instances to generate an additional 1,000 instances using each technique. Subsequently, we collect 10,000 training samples by running a strong branching expert again on these instances. In combination with the original 1,000 samples, we have a comprehensive enriched training dataset consisting of 11,000 samples. Thus, the three generation techniques lead to the following approaches: *GNN+Bowly*, *GNN+G2MILP* $\eta = x$, *GNN+MILP-StuDio* $\eta = x$, where $\eta = x$ implies that we set the modification ratio to be $x$. Additionally, we compare our method with a GNN model trained on 11,000 samples collected from 1,100 original instances (*GNN-11000*). This comparison allows us to demonstrate that the improvement achieved by our approach is not only due to an increase in the number of training samples but also the high quality of the generated instances. To provide a comprehensive evaluation, we also compare the solving performance of our method with that of the SCIP solver [6] (*SCIP*), which serves as our backend solver for comparison purposes.

**Experiment Results**  In this section, we conduct a comprehensive comparison of imitation accuracy and solving performance. As discussed in Appendix E.1, imitation accuracy serves as a crucial

Table 5: Comparison of branching accuracy on the testing datasets. 'Trivial samples' implies that the computational hardness of the generated instances is so low that the solver does not need to perform branching to solve them. As a result, we are unable to collect any branching samples for these trivial instances. We mark **the best** performance in bold among the methods using the generation technique. We can see that MILP-StuDio can consistently improve the branching accuracy of the learned models.

| | FA | IP |
|---|---|---|
| GNN-11000 | 0.621 | 0.780 |
| GNN | 0.545 | 0.640 |
| GNN+Bowly | Trivial samples | 0.358 |
| GNN+G2MILP $\eta = 0.01$ | 0.535 | 0.765 |
| GNN+G2MILP $\eta = 0.05$ | 0.545 | 0.705 |
| GNN+G2MILP $\eta = 0.10$ | 0.560 | 0.705 |
| GNN+MILP-StuDio $\eta = 0.01$ | 0.585 | **0.800** |
| GNN+MILP-StuDio $\eta = 0.05$ | **0.605** | 0.790 |
| GNN+MILP-StuDio $\eta = 0.10$ | 0.560 | 0.740 |

Table 6: Comparison of solving performance in learning-to-branch between our approach and baseline methods, under a $1,000$-second time limit. 'Obj' represents the objective values achieved by different methods and 'Time' denotes the average time used to find the solutions. '↓' indicates that lower is better. We mark **the best** values in bold. 'Trivial samples' implies that the computational hardness of the generated instances is so low that the solver does not need to perform branching to solve them. As a result, we are unable to collect any branching samples for these trivial instances.

| | FA | | IP | |
|---|---|---|---|---|
| | Obj ↓ | Time ↓ | Obj ↓ | Time ↓ |
| SCIP | 17865.38 | 58.18 | 22.03 | 1000.00 |
| GNN-1000 | 17865.38 | 51.63 | 20.72 | 1000.00 |
| GNN | 17865.38 | 55.06 | 21.04 | 1000.00 |
| GNN+Bowly | Trivial samples | | 22.58 | 1000.00 |
| GNN+G2MILP $\eta = 0.01$ | 17865.38 | 58.81 | 21.09 | 1000.00 |
| GNN+G2MILP $\eta = 0.05$ | 17865.38 | 61.33 | 22.11 | 1000.00 |
| GNN+G2MILP $\eta = 0.10$ | 17865.38 | 59.64 | 20.93 | 1000.00 |
| GNN+MILP-StuDio $\eta = 0.01$ | 17865.38 | 48.28 | **20.03** | **1000.00** |
| GNN+MILP-StuDio $\eta = 0.05$ | **17865.38** | **47.58** | 21.00 | 1000.00 |
| GNN+MILP-StuDio $\eta = 0.10$ | 17865.38 | 53.70 | 20.39 | 1000.00 |

metric for evaluating the performance of the GNN model. Thus, we first compare our methods with other GNN-based baselines, including GNN, GNN-11000, GNN+Bowly, and GNN+G2MILP regarding imitation accuracy. The corresponding results are presented in Table 5, where we observe that MILP-StuDio yields the most significant improvement in imitation accuracy. Interestingly, we find that the instances generated by Bowly in FA are computationally trivial, such that the solver does not need to perform any branching operations to solve them. Consequently, we are unable to collect additional branching samples from these trivial instances to further train the GNN model.

Furthermore, we evaluate the solving performance of the different baselines, as summarized in Table 6. Notably, GNN+MILP-StuDio outperforms all the generation-technique-enhanced baselines and achieves comparable solving performance of GNN-11000. This suggests that incorporating the MILP-StuDio method can boost the performance of the GNN branching policy. We also find that the generated data from MILP-StuDio with different modification ratios can be beneficial for training the GNN model while using instances generated by the G2MILP method may potentially disrupt the training of the GNN model. These findings substantiate the effectiveness of MILP-StuDio in enhancing the GNN branching policy, both in terms of imitation accuracy and solving performance.

Table 7: Structural similarity scores between the generated and original instances in FA. We compare the effect of different generation operators and modification ratios. We find that the mix-up operator can generate the most realistic instances.

|  | $\eta = 0.01$ | $\eta = 0.05$ | $\eta = 0.10$ |
|---|---|---|---|
| G2MILP | 0.358 | 0.092 | 0.091 |
| MILP-StuDio (red) | 0.541 | 0.515 | 0.474 |
| MILP-StuDio (mix) | 0.907 | 0.908 | 0.907 |
| MILP-StuDio (exp) | 0.543 | 0.517 | 0.475 |

Table 8: Average solving time (s) and feasibility of instances solved by Gurobi. $\eta$ is the masking ratio. Numbers in the parentheses are feasible ratios in the instances. The instances generated by the G2MILP are found to have an extremely low feasible ratio.

|  | $\eta = 0.01$ | $\eta = 0.05$ | $\eta = 0.10$ |
|---|---|---|---|
| G2MILP | 0.02 (1.7%) | 0.01 (1.8%) | 0.01 (2.0%) |
| MILP-StuDio (red) | 4.29 (100.0%) | 4.46 (100.0%) | 4.26 (100.0%) |
| MILP-StuDio (mix) | 5.35 (100.0%) | 5.00 (100.0%) | 4.26 (100.0%) |
| MILP-StuDio (exp) | 5.18 (100.0%) | 5.30 (100.0%) | 5.42 (100.0%) |

## F.2 More Results on Different Generation Operators and Modification Ratios

In this part, we investigate the influence of different generation operators and modification ratios of MILP-StuDio. 'MILP-StuDio (Optor) $\eta = x$' denotes the generation approach that uses the operator Optor ('red' means reduction, 'mix' represents mix-up, and 'exp' denotes expansion) and modification ratio $\eta = x$.

**Influence on the Similarity between the Generated and the Original Instances**  We conduct experiments on the FA benchmark. We first compute the graph distributional similarity score for the 100 original instances and 1,000 instances generated using different operators and modification ratios. The results presented in Table 7 suggest that the choice of generation operator and modification ratio can impact the values of the similarity scores. (1) MILP-StuDio consistently outperforms G2MILP in similarity scores. (2) Among the three operators, mix-up achieves the highest similarity scores, with values exceeding 0.8 across the modification ratios. Although the reduction and expansion operators yield lower similarity scores, they are still able to achieve comparable performance in the downstream tasks, as shown in the following paragraph. This finding indicates that there is not necessarily a positive correlation between the structural similarity scores and the benefits for the downstream tasks.

We also evaluate the computational properties of the generated instances. The results in Table 8 show that the three operators are able to preserve the computational hardness of the original instances. Importantly, all the operators succeed in generating feasible instances, which is a crucial requirement for the downstream tasks. In contrast, the instances generated by the G2MILP approach are found to have an extremely low feasible ratio.

Moreover, as the modification ratio increases, we observe a decrease in the similarity scores. Considering the good computational properties we have discussed, this finding suggests that the increasing modification ratios lead to the generation of more novel instances. This observation indicates that MILP-StuDio is able to generate instances increasingly distinct from the original data, while still preserving the computational hardness and feasibility properties.

**Influence of Generation Operators on the Performance of Predict-and-Search**  We investigate the influence of instances generated with different operators on the performance of Predict-and-Search (PS). Specifically, we conduct experiments on the FA and IP benchmarks, using 1,000 generated instances based on 100 original instances and different operators. For comparison, we also report the results of PS+MILP-StuDio $\eta = 0.05$. In PS+MILP-StuDio $\eta = 0.05$, we use the three

Table 9: Comparison of prediction loss on the testing datasets for different generation operators. We set the modification ratio $\eta = 0.05$. We mark **the best** values in bold.

|  | FA | IP |
|---|---|---|
| PS | 20.45 | 341.33 |
| PS+MILP-StuDio $\eta = 0.05$ | 18.25 | 341.33 |
| PS+MILP-StuDio (red) $\eta = 0.05$ | **15.53** | **341.33** |
| PS+MILP-StuDio (mix) $\eta = 0.05$ | 20.44 | 341.33 |
| PS+MILP-StuDio (exp) $\eta = 0.05$ | 20.56 | 341.33 |

Table 10: Comparison of solving performance in PS between different operators of MILP-StuDio, under a $1,000$-second time limit. In the IP benchmark, all the approaches reach the time limit of 1,000s, thus we do not consider the Time metric. '↓' indicates that lower is better.

|  | FA (BKS 17865.38) | | | IP (BKS 11.16) | |
|---|---|---|---|---|---|
|  | Obj ↓ | gap$_{abs}$ ↓ | Time ↓ | Obj ↓ | gap$_{abs}$ ↓ |
| PS | 17865.38 | 0.00 | 7.19 | 11.40 | 0.24 |
| PS+MILP-StuDio $\eta = 0.05$ | 17865.38 | 0.00 | 7.07 | **11.29** | **0.13** |
| PS+MILP-StuDio (red) $\eta = 0.05$ | 17865.38 | 0.00 | 7.01 | 11.38 | 0.22 |
| PS+MILP-StuDio (mix) $\eta = 0.05$ | **17865.38** | **0.00** | **6.95** | 11.32 | 0.16 |
| PS+MILP-StuDio (exp) $\eta = 0.05$ | 17865.38 | 0.00 | 6.96 | 11.37 | 0.21 |

operators (red, mix, and exp) to generate 1,000 instances (333, 334, and 333 using the three operators, respectively). The results are presented in Tables 9 and 10, which compare the prediction loss and solving performance on the testing datasets for the different methods. The key findings are as follows. (1) All three generation operators are beneficial for the performance of the PS algorithm. (2) The PS algorithm is not overly sensitive to the choice of generation operator (3) The most beneficial operator may differ across different benchmarks.

**Influence of Modification Ratio on the Performance of Predict-and-Search** We investigate how the modification ratio impacts the performance of PS. Specifically, we conduct experiments on the FA and IP benchmarks, generating 1,000 instances using 100 original instances with different modification ratios $\{0.01, 0.05, 0.10\}$ for both the MILP-StuDio and G2MILP approaches. In PS+MILP-StuDio, we use the three operators (red, mix, and exp) to generate one-third of the 1,000 instances, respectively. The results are presented in Tables 11 and 12, in which we compare the prediction loss and solving performance on the testing datasets for the different methods. Tables 11 and 12 showcase the following results. (1) MILP-StuDio consistently outperforms G2MILP across modification ratios. (2) A smaller modification ratio in MILP-StuDio can lead to a slightly better performance of PS. (3) MILP-StuDio is not sensitive to the masking ratio in general.

### F.3 Enhancing the Traditional Solvers via Hyperparameter Tuning

In our Gurobi hyperparameter tuning experiment, We employ the Bayesian optimization framework provided by the HEBO package [59]. Each tuning process involves 100 trials, where different hyperparameter configurations are sampled and evaluated. To strike a balance between tuning efficiency and effectiveness, we focus on optimizing eight key hyperparameters: Heuristics, MIPFocus, VarBranch, BranchDir, Presolve, PrePasses, Cuts, and Method. We list and briefly introduce the key hyperparameters in Table 13.

We utilize the Bayesian optimization package HEBO [59] to perform hyperparameter tuning for the Gurobi solver. We start by using the original dataset consisting of 20 FA instances and generate an additional 200 instances using the MILP-StuDio framework with a modification ratio of $\eta = 0.05$. This enriched dataset is then used to run the hyperparameter optimization process, where we perform 100 trials to search for the best parameter configuration. Finally, we evaluate the performance of the tuned Gurobi solver on a testing dataset of 50 instances. We denote the default Gurobi as Gurobi, the Gurobi tuned on the 20 original instances as Gurobi-tuned, and the Gurobi tuned with additional instances generated by Bowly, G2MILP, and MILP-StuDio as tuned-Bowly, tuned-G2MILP, and tuned-MILP-StuDio, respectively. Experiment results in Table 14 demonstrate that Gurobi tuned with

Table 11: Comparison of prediction loss using different modification ratios on the testing datasets. The notation 'infeasible' implies that a majority of the generated instances are infeasible and thus cannot be used as the training data for PS. We mark **the best** values in bold.

|  | FA | IP |
|---|---|---|
| PS | 19.45 | 341.33 |
| PS+G2MILP $\eta = 0.01$ | Infeasible | 348.90 |
| PS+MILP-StuDio $\eta = 0.01$ | 18.52 | 341.33 |
| PS+G2MILP $\eta = 0.05$ | Infeasible | 363.76 |
| PS+MILP-StuDio $\eta = 0.05$ | **18.25** | **341.33** |
| PS+G2MILP $\eta = 0.10$ | Infeasible | 348.91 |
| PS+MILP-StuDio $\eta = 0.10$ | 19.39 | 341.33 |

Table 12: Comparison of solving performance in PS with different modification ratios of MILP-StuDio and G2MILP, under a 1,000-second time limit. In the IP benchmark, all the approaches reach the time limit of 1,000s, thus we do not consider the Time metric. The notation 'Infeasible' implies that a majority of the generated instances are infeasible and thus cannot be used as the training data for PS. '↓' indicates that lower is better. We mark **the best** values in bold.

|  | FA (BKS 17865.38) | | | IP (BKS 11.16) | |
|---|---|---|---|---|---|
|  | Obj ↓ | gap$_{abs}$ ↓ | Time ↓ | Obj ↑ | gap$_{abs}$ ↓ |
| PS | 17865.38 | 0.00 | 7.19 | 11.40 | 0.24 |
| PS+G2MILP $\eta = 0.01$ | Infeasible | | | 11.57 | 0.41 |
| PS+MILP-StuDio $\eta = 0.01$ | 17865.38 | 0.00 | 7.08 | **11.22** | **0.06** |
| PS+G2MILP $\eta = 0.05$ | Infeasible | | | 11.55 | 0.39 |
| PS+MILP-StuDio $\eta = 0.05$ | **17865.38** | **0.00** | **7.07** | 11.29 | 0.13 |
| PS+G2MILP $\eta = 0.10$ | Infeasible | | | 11.62 | 0.46 |
| PS+MILP-StuDio $\eta = 0.10$ | 17865.38 | 0.00 | 7.11 | 11.53 | 0.37 |

Table 13: Selected hyperparameters of Gurobi.

| Hyperparameter | Type | Min | Max | Description |
|---|---|---|---|---|
| Heuristics | double | 0 | 1 | Time spent in feasibility heuristics. |
| MIPFocus | integer | 0 | 3 | Set the focus of the MIP solver. |
| VarBranch | integer | -1 | 3 | Branch variable selection strategy. |
| BranchDir | integer | -1 | 1 | Preferred branch direction. |
| Presolve | integer | -1 | 2 | Controls the presolve level. |
| PrePasses | integer | -1 | 20 | Presolve pass limit. |
| Cuts | integer | -1 | 3 | Global cut generation control. |
| Method | integer | -1 | 5 | Algorithm used to solve continuous models. |

additional instances generated by the MILP-StuDio framework (tuned-MILP-StuDio) consistently outperforms the other baseline methods.

## F.4 More Results on Non-Structured MILPs

While MILP-StuDio is primarily designed to handle MILP instances with block structures in their CCMs, it is also important to evaluate its performance on wider classes of MILP problems, including those that may not exhibit such structured characteristics. Evaluating MILP-StuDio's capabilities in this broader context will highlight its potential for broader applications.

To extend our method to general MILP instances, the framework remains unchanged and we just need to adjust the block decomposition and partition Algorithms 1 and 2. Specifically, the constraint

Table 14: Comparison of solving performance in Gurobi hyperparameter tuning. We mark **the best** performance in bold.

|  | FA | |
| --- | --- | --- |
|  | $\text{gap}_{abs} \downarrow$ | Time $\downarrow$ |
| Gurobi | 0.00 | 6.06 |
| Gurobi-tuned | 0.00 | 6.04 |
| tuned-Bowly | 0.00 | 5.96 |
| tuned-G2MILP | Infeasible | |
| tuned-MILP-StuDio | **0.00** | **5.83** |

and variable classification Algorithm 1 is tailored to general MILPs. The graph community reflects the connection of the constraints and variables, which can serve as a generalization of blocks. Block partition and community discovery are both graph node clustering. Thus, we cluster the constraints and variables according to the community and classification results to form generalized 'blocks'.

To this end, we provide more results on the SC (set covering) and MIS (maximum independent set) datasets (two popular benchmarks with non-structural instances from [8]) and further investigate the improvement of the ML solvers. We use 100 original instances, following the generation algorithm described in [8] and [15]), and generate 1,000 new instances using G2MILP and MILP-StuDio. The results demonstrate the outstanding performance of MILP-StuDio on general MILP instances without block structures. (1) Tables 15 and 17 show that MILP-StuDio can even outperform existing generation methods in terms of the similarity score and feasibility ratio. (2) Tables 15 and 17 show that MILP-StuDio better preserves instances' computational hardness with closer solving time to the original instances. (3) Tables 16 and 18 show that MILP-StuDio leads to the greatest improvement for the PS solver.

Table 15: The similarity score, computational hardness, and feasibility ratio between the original and generated instances on the non-structural large-scale Setcover benchmark (larger than that in Appendix E.4). We set the solving time limit 300s and $\eta = 0.05$. We mark **the best** performance in bold.

|  | Original | Bowly | G2MILP | MILP-StuDio |
| --- | --- | --- | --- | --- |
| Similarity | 1.00 | 0.231 | 0.763 | **0.766** |
| Solving Time | 300.0 | 3.72 | 300.0 | **300.0** |
| Feasibility Ratio | 100% | 100% | 100% | **100%** |

Table 16: The performance of the PS solver trained by instances generated by different methods on the non-structural Setcover benchmark. We set the solving time limit 300s and $\eta = 0.05$. We mark **the best** performance in bold.

|  | Gurobi | PS | Bowly | G2MILP | MILP-StuDio |
| --- | --- | --- | --- | --- | --- |
| Time | 300.0 | 300.0 | 300.0 | **300.0** | **300.0** |
| Obj | **123.80** | 123.95 | 124.00 | **123.80** | **123.80** |
| $\text{gap}_{abs}$ | **0.21** | 0.22 | 0.22 | **0.21** | **0.21** |

Table 17: The similarity score, computational hardness, and feasibility ratio between the original and generated instances on the non-structural MIS benchmark. We set the solving time limit 300s and $\eta = 0.05$. We mark **the best** performance in bold.

|  | Original | Bowly | G2MILP | MILP-StuDio |
|---|---|---|---|---|
| Similarity | 1.00 | 0.182 | **0.921** | 0.919 |
| Solving Time | 300.0 | 1.38 | **300.0** | 300.0 |
| Feasibility Ratio | 100% | 100% | **100%** | 100% |

Table 18: The performance of the PS solver trained by instances generated by different methods on the non-structural MIS benchmark. We set the solving time limit 300s and $\eta = 0.05$. We mark **the best** performance in bold.

|  | Gurobi | PS | Bowly | G2MILP | MILP-StuDio |
|---|---|---|---|---|---|
| Time | 119.74 | 36.85 | 42.92 | 21.39 | **11.33** |
| Obj | 683.75 | 683.75 | 683.75 | 683.75 | **683.75** |
| $gap_{abs}$ | 0.00 | 0.00 | 0.00 | 0.00 | **0.00** |

## F.5 More Results on Real-world Industrial Dataset

To further demonstrate the effectiveness in real-world applications, we also conduct experiments on a real-world scheduling dataset from an anonymous enterprise, which is one of the largest global commercial technology enterprises. The instances do not present clear block structures. The results in Table 19 show that the extended framework generalizes well on the general MILP datasets and has promising potential for real-world applications. The dataset contains 36 training and 12 testing instances. The few training instances reflect the data inavailability problem in real-world applications. We use different data generation methods to enhance the performance of the MLPS solver and list the solving time, objective value, and node number in Table 19 as follows. MILP-StuDio outperforms other baselines in this dataset, highlighting its strong performance and applicability.

Table 19: The results in the real-world dataset.

|  | Time (1000s time limit) | | | Obj | | | Node Number | | |
|---|---|---|---|---|---|---|---|---|---|
|  | PS | PS+G2MILP | PS+MILP-StuDio | PS | PS+G2MILP | PS+MILP-StuDio | PS | PS+G2MILP | PS+MILP-StuDio |
| instance1 | 1509652.00 | 1509652.00 | 1509652.00 | 144.21 | 146.67 | 131.80 | 38767.00 | 37647.00 | 39513.00 |
| instance2 | 205197.00 | 205197.00 | 205197.00 | 5.83 | 9.51 | 10.11 | 723.00 | 746.00 | 1047.00 |
| instance3 | 7483.00 | 7483.00 | 7483.00 | 29.50 | 32.68 | 38.49 | 555.00 | 751.00 | 901.00 |
| instance4 | 447781.00 | 384454.99 | 318675.99 | 1000.00 | 1000.00 | 1000.00 | 16545.00 | 6201.00 | 5186.00 |
| instance5 | 1465601.00 | 1465601.00 | 1465601.00 | 186.85 | 81.67 | 63.55 | 24317.00 | 11481.00 | 9732.0 |
| instance6 | 1293554.00 | 1293554.00 | 1293554.00 | 38.65 | 25.41 | 32.45 | 3471.00 | 2357.00 | 3002.00 |
| instance7 | 1293554.00 | 1293554.00 | 1293554.00 | 38.96 | 24.82 | 31.76 | 3471.00 | 2357.00 | 3002.00 |
| instance8 | 612151.00 | 612151.00 | 612151.00 | 0.18 | 0.35 | 0.28 | 13.00 | 55.00 | 13.00 |
| instance9 | 1578141.00 | 1578141.00 | 1578141.00 | 28.53 | 26.74 | 23.61 | 3083.00 | 3207.00 | 3150.00 |
| instance10 | 1149250.00 | 1149250.00 | 1149250.00 | 7.33 | 8.89 | 8.05 | 1.00 | 154.00 | 1.00 |
| instance11 | 1030555.00 | 1030555.00 | 1030555.00 | 0.27 | 0.35 | 0.34 | 1.00 | 1.00 | 1.00 |
| instance12 | 1216445.00 | 1216445.00 | 1216445.00 | 23.64 | 23.10 | 13.83 | 1384.00 | 1384.00 | 1.00 |
| Average | 984113.66 | 978836.50 | **973354.92** | 125.35 | 115.02 | **112.86** | 7694.25 | 5528.42 | **5462.42** |

## G Visualizations of CCMs

To provide further insights into the characteristics of the instances generated by MILP-StuDio, we visualize the Constraint Coefficient Matrices (CCMs) of both the original and generated instances in Figure 7-10. The CCM visualization is a powerful tool to understand the structural properties of MILP instances, as it captures the patterns and relationships between the constraints and variables. By comparing the CCMs of the original and generated instances, we can understand how well MILP-StuDio is able to preserve the key structural characteristics of the input problems.

As shown in Figure 7-10, MILP-StuDio is able to successfully maintain the block structures observed in the original instances across the various benchmark problems. This indicates that the generated instances share similar underlying problem structures with the original data, which is a desirable property. This ability to preserve the structural properties of MILP instances is crucial for the effective training and evaluation of machine learning-based MILP solvers. In contrast, Bowly struggles to

capture the intricate CCM structures, while G2MILP introduces additional noise that disrupts the block structures in the generated CCMs.

# H  Implementation Details

## H.1  Implementation of Predict-and-Search

For model structure, the PS models used in this paper align with those outlined in the original papers [10]. We use the code in `https://github.com/sribdcn/Predict-and-Search` MILP method to implement PS. For the PS predictor, we leverage a graph neural network comprising four half-convolution layers. We conducted all the experiments on a single machine with NVidia GeForce GTX 3090 GPUs and Intel(R) Xeon(R) E5-2667 V4CPUs 3.20GHz.

In the training process of MILP-StuDio, we set the initial learning rate to be 0.001 and the training epoch to be 1000 with early stopping. In addition, the partial solution size parameter $(k_0, k_1)$ and neighborhood parameter $\Delta$ are two important parameters in PS. The partial solution size parameter $(k_0, k_1, \Delta)$ represents the numbers of variables fixed with values 0 and 1 in a partial solution. The neighborhood parameter $\Delta$ defines the radius of the searching neighborhood. We list these two parameters used in our experiments in Table 20.

Table 20: The partial solution size parameter $(k_0, k_1, \Delta)$ and neighborhood parameter $\Delta$.

| Benchmark | CA | FA | IP | WA |
|---|---|---|---|---|
| $(k_0, k_1, \Delta)$ | (300,0,10) | (20,0,10) | (50,5,10) | (0,600,5) |

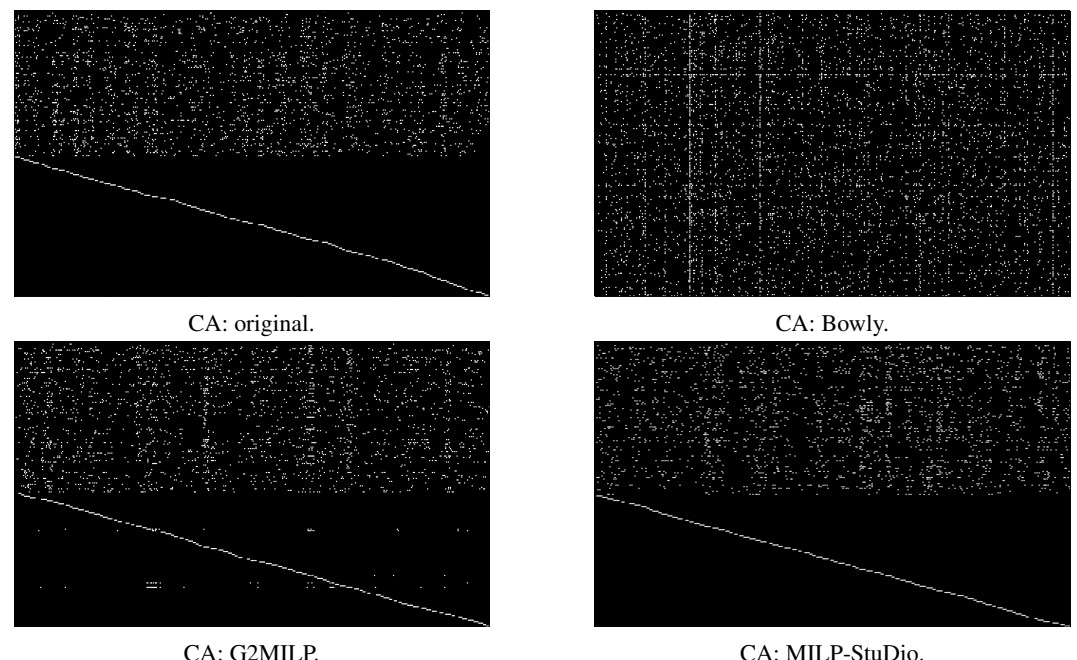

CA: original.                    CA: Bowly.

CA: G2MILP.                    CA: MILP-StuDio.

Figure 7: The visualization of the CCMs of original and generated instances from CA.

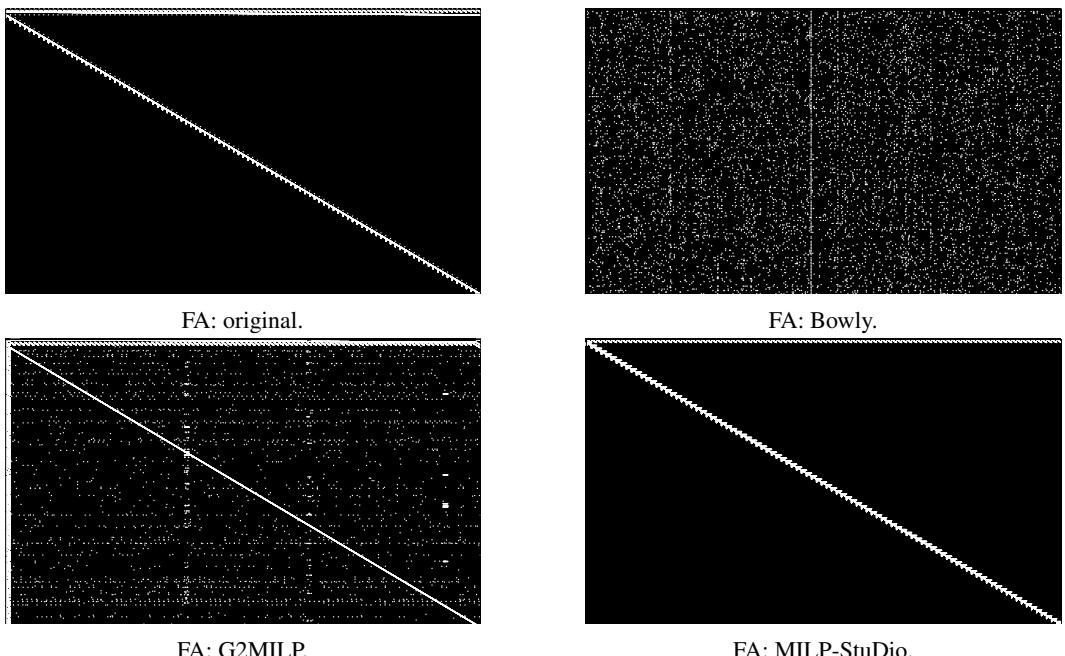

FA: original.                    FA: Bowly.

FA: G2MILP.                    FA: MILP-StuDio.

Figure 8: The visualization of the CCMs of original and generated instances from FA.

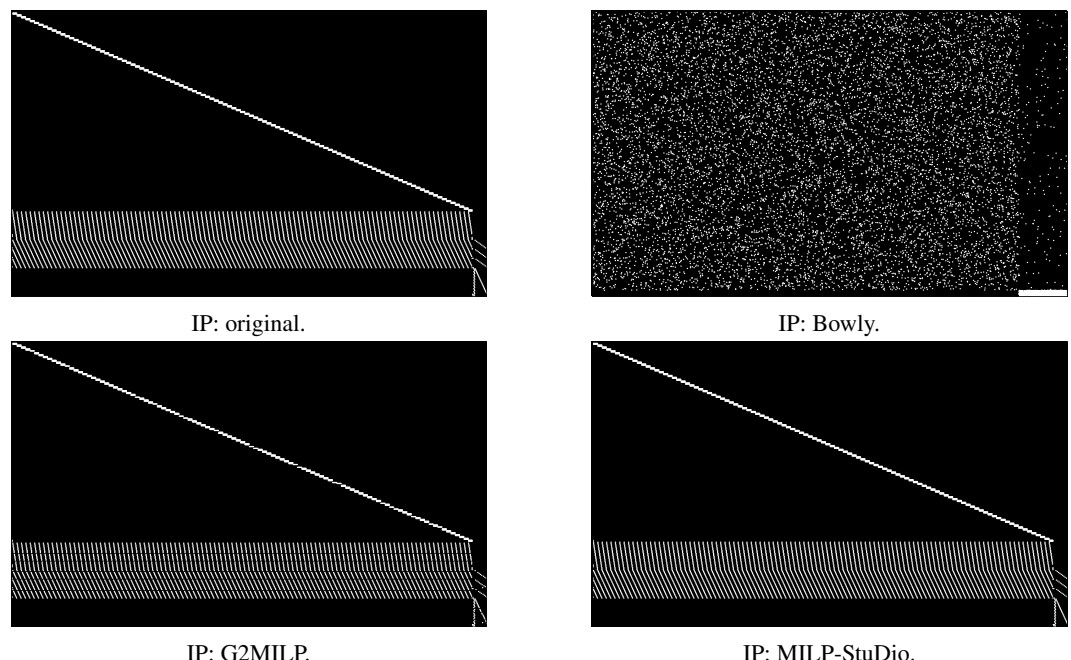

Figure 9: The visualization of the CCMs of original and generated instances from IP.

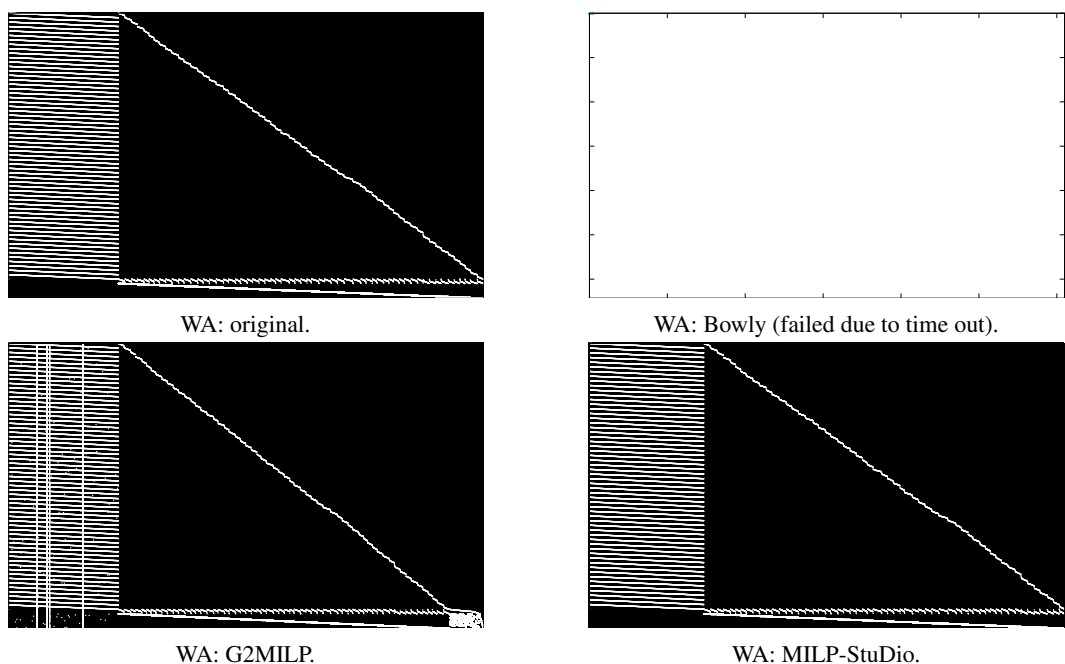

Figure 10: The visualization of the CCMs of original and generated instances from WA.

## H.2 Implementation of Classification Algorithm for Constraints and Variables

In Section 4.1, we classified the constraints into M-Cons, B-Cons, and DB-Cons, and the variables into Bl-Vars and Bd-Vars. As shown in Equation (4), we show the constraint and variable classification results for the bordered block-diagonal and doubly bordered block-diagonal structures.

$$
\left.\begin{pmatrix} D_1 & & & \\ & D_2 & & \\ & & \ddots & \\ & & & D_k \\ B_1 & B_2 & \cdots & B_k \end{pmatrix}\begin{array}{l}\\ \\ \left.\vphantom{\begin{matrix}D_1\\D_2\\\ddots\\D_k\end{matrix}}\right\}\text{B-Cons}\\ \\ \left.\vphantom{B_k}\right\}\text{M-Cons}\end{array}\qquad \left.\begin{pmatrix} D_1 & & & & F_1 \\ & D_2 & & & F_2 \\ & & \ddots & & \vdots \\ & & & D_k & F_k \\ B_1 & B_2 & \cdots & B_k & C \end{pmatrix}\begin{array}{l}\\ \left.\vphantom{\begin{matrix}D_1\\D_2\\\ddots\\F_k\end{matrix}}\right\}\text{B-Cons}\\ \\ \left.\vphantom{C}\right\}\text{M-Cons}\end{array}\right. \tag{4}
$$

$$\underbrace{\qquad\qquad\qquad}_{\text{Bl-Vars}}\qquad\qquad\underbrace{\qquad\qquad}_{\text{Bl-Vars}}\quad\text{Bd-Vars}$$

Identifying these different types of constraints and variables in a CCM can aid the (1) variable partition process during block decomposition and (2) block manipulation process during generation. Considering the CCM, we denote the following attributes for each row (constraint) and column (variable). For a row in CCM $A \in \mathbb{R}^{m \times n}$, $x_{\max}$ is the maximum $x$-coordinate of nonzero entries, $x_{\min}$ is the minimum $x$-coordinate of nonzero entries, and $h_x$ is the number of nonzero entries. Similarly, given a column in CCM, we denote the maximum $y$-coordinate of the nonzero entries by $y_{\max}$, the minimum $y$-coordinate of the nonzero entries by $y_{\min}$, and the number of nonzero entries by $h_y$. we first give the following observations.

- The nonzero entries in an M-Con often have a wide range of $x$-indices, i.e., large $x_{\max} - x_{\min}$ and high standard deviation of $x$.
- The nonzero entries in a B-Con often have a narrow range of $x$-indices, i.e., small $x_{\max} - x_{\min}$ and small standard deviation of $x$.
- For a Bd-Var, the nonzero entries in the corresponding column of CCM have a wide range of $y$-indices and high density (the proportion of nonzero entries in the column), i.e., large $y_{\max} - y_{\min}$ and $h_y/m$.
- DB-Cons are defined as constraints containing Bd-Vars.

Using these observations, we define vector features for each row and column in the CCM. The row features include the standard deviation (`row_feat[0]`), the proportion of nonzeros (density, `row_feat[1]`), and ranges of the $x$-coordinates (`row_feat[2]`). For a column in CCM, we consider the range (`col_feat[0]`) and density (`col_feat[1]`) of the nonzero entries. The pseudo-code of the classification algorithm is presented in Algorithm 1. For the hyperparameter used in this paper, we set $\phi_1 = 0.75$, $\phi_2 = 0.75$, $\phi_3 = 0.5$, $\phi_4 = 0.2$, and $\phi_5 = 0.5$, and set $DB =$True only in WA.

## H.3 Implementation of Partition Algorithm for Block Units

In Section 4.2, we use GCG to reorder the rows and columns in CCMs and obtain an initial variable partition result from GCG. However, GCG sometimes cannot partition the variables well, such as the situation in CA where GCG fails to identify the blocks since the blocks in CA contain different numbers of variables. Thus, we need to refine the partition to obtain better ones. Specifically, we find that many blocks in CCMs contain regular lines (mainly horizontal and diagonal lines). Thus, we define a line detection for block identification. Splitting these lines by columns we can get the block units from the CCMs. To get block units by lines from the reordered CCM image, we design the following partition algorithm in Algorithm 2. The algorithm iterates over the points in the image and finds the endpoint in the columns of the lines mentioned above. With all the endpoints in columns, we can split one image into block units in columns. To keep the algorithm concise, we introduce two criteria, each representing a different detection method.

- Criterion 1: horizontal line detection. This criterion can help to locate the endpoint of horizontal lines in the image precisely.
- Criterion 2: diagonal line detection. This criterion can also help to locate the endpoint of diagonal lines in the image precisely.

We set $\zeta = 4$ for CA and $\zeta = 3$ for other benchmarks.

---

**Algorithm 1:** Classification algorithm for constraints and variables in CCMs

---

**Input:** The CCM $\boldsymbol{A} \in \mathbb{R}^{m \times n}$ to be analyzed, hyperparameter $\phi_1$, $\phi_2$, $\phi_3$, $\phi_4$, and $\phi_5$, the binary controller $DB$ to determine whether identify DB-Vars

**Output:** The classification results for each row and column: B-Conslist, M-Conslist, DB-Conslist, Bl-Varslist, and Bd-Varslist

**1** `Initialize:` set B-Conslist, M-Conslist, DB-Conslist, Bl-Varslist, and Bd-Varslist to be empty
**2** Compute the column features and row features of $\boldsymbol{A}$ and normalize them
**3** **if** $DB =$*True* **then**
**4**      **for** *j in* $\{1, \cdots, n\}$ **do**
**5**          **if** *the features of the j-th variable* `col_feat[0]` $> \phi_1$ *and* `col_feat[1]` $> \phi_2$ **then**
**6**             Append $j$ to Bd-Varslist
**7**          **end**
**8**          **else**
**9**             Append $j$ to Bl-Varslist
**10**          **end**
**11**      **end**
**12** **end**
**13** Append the indices of constraints that contain variables in Bl-Varslist into DB-Conslist
**14** **for** *i in* $\{1, \cdots, m\}$ **do**
**15**      **if** *the i-th constraint is not in DB-Cons and the features of the i-th constraint* `row_feat[0]` $> \phi_3$, `row_feat[1]` $> \phi_4$ *and* `row_feat[2]` $> \phi_5$ **then**
**16**          Append $i$ to M-Conslist
**17**      **end**
**18**      **else**
**19**          Append $i$ to B-Conslist
**20**      **end**
**21** **end**
**22** **return** B-Conslist, M-Conslist, DB-Conslist, Bl-Varslist, and Bd-Varslist

---

---

**Algorithm 2:** Partition algorithm for variables in CCMs

---

**Input:** The image representation $\tilde{\boldsymbol{A}} \in \mathbb{R}^{m \times n}$ of reordered CCM to be analyzed

**Output:** The partition results for the columns

**1** `Initialize:` set Partitionlist to be empty, cut-off point p and q, detection horizon $\zeta$.
**2** **Criterion 1:** $\tilde{\boldsymbol{A}}[i-1][j-1] = \tilde{\boldsymbol{A}}[i-2][j-2] = \cdots = \tilde{\boldsymbol{A}}[i-\zeta][j-\zeta] = 255$ and $\tilde{\boldsymbol{A}}[i+1][j+1] = 0$.
**3** **Criterion 2:** $\tilde{\boldsymbol{A}}[i-1][j] = \tilde{\boldsymbol{A}}[i-2][j] = \cdots = \tilde{\boldsymbol{A}}[i-\zeta][j] = 255$ and $\tilde{\boldsymbol{A}}[i+1][j] = 0$.
**4** **for** *j in* $\{1, \cdots, n\}$ **do**
**5**      **for** *i in* $\{1, \cdots, m\}$ **do**
**6**          **if** $\tilde{\boldsymbol{A}}[i][j] = 255$ **then**
**7**             **if** *Criterion 1 or Criterion 2 is satisfied* **then**
**8**                 $q = j$
**9**                 Append $[p:q]$ to Partitionlist
**10**                 $p = q$
**11**                 **break**
**12**             **end**
**13**          **end**
**14**      **end**
**15** **end**
**16** **return** Partitionlist

---

## H.4 Details on Block Manipulation

**Block manipulation** We leverage the classification results for constraints and variables to aid the block manipulation process, in which the results can help us process more complex structures of CCMs beyond the three basic ones. For example, we can process the following structures in Equation 5, which is the combination of DB-Cons, M-Cons, and B-Cons.

$$
\begin{pmatrix}
\boldsymbol{D}_1 & & & & \boldsymbol{F}_1 \\
& \boldsymbol{D}_2 & & & \boldsymbol{F}_2 \\
& & \ddots & & \vdots \\
& & & \boldsymbol{D}_k & \boldsymbol{F}_k \\
\boldsymbol{B}_1 & \boldsymbol{B}_2 & \cdots & \boldsymbol{B}_k & \boldsymbol{C} \\
\tilde{\boldsymbol{D}}_1 & & & & \\
& \tilde{\boldsymbol{D}}_2 & & & \\
& & \ddots & & \\
& & & \tilde{\boldsymbol{D}}_k &
\end{pmatrix}
\tag{5}
$$

Different types of constraints and variables have different manipulation methods. For example, when we apply the reduction and expansion operators, the manipulation of Bl-Vars can change the variable numbers, and the manipulation of B-Cons and DB-Cons can change the constraint numbers.

For the mix-up and expansion operators, the number of the introduced M-Cons $m_1$ may mismatch that in the original instance $m_2$. We define a successful matching if $m_1 \geq m_2$ If $m_1 = m_2$, the operators can be performed well. Otherwise, we drop the last $m_1 - m_2$ M-Cons in the Block unit, such that the M-Cons can be perfectly matched.

**Coefficient refinement** In addition to the block structures, we have observed the presence of specific patterns in the coefficient values of real-world MILP instances. These patterns contribute to the overall problem structure and are worth considering during the instance generation process. For instance, as shown in Equation (6), we write two CCMs from different instances. In certain MILP instances (like FA), the nonzero coefficients within the B-Cons (rows of $\boldsymbol{D}$) remain consistent across the block units within a given instance (marked in the same color, blue or red, in the same instance), but may differ across different instances (marked in different colors). However, when applying mix-up or expansion operators to generate new instances, the introduced constraint coefficients from other instances can potentially disrupt this inherent coherence.

To address this issue, we have incorporated a constraint refinement component that focuses on preserving the distribution of coefficient values in such scenarios. Specifically, we define a "non-trivial constraint" as a constraint that contains values other than 0, -1, and 1. For each MILP instance, we identify the non-trivial constraints in the block units. For the $k$-th non-trivial constraint $\mathbf{a}_k$ in the block unit, we then compute the mean $\mu_k^{(i)}$ and variance $\sigma_k^{(i)}$ across the constraint coefficients and block units in this instance. During the instance generation process, whenever the refinement component is triggered (e.g., after mix-up or expansion operations), it samples the new constraint coefficients for the introduced blocks from a Gaussian distribution $\mathcal{N}(\mu_k^{(i)}, \sigma_k^{(i)})$. This ensures that the generated instances maintain a similar distribution of non-trivial coefficients as observed in the original instances. The pseudo code of coefficient refinement is in Algorithm 3 and we activate this component in FA and IP.

$$
\begin{pmatrix}
\textcolor{blue}{\boldsymbol{D}} & & & \\
& \textcolor{blue}{\boldsymbol{D}} & & \\
& & \ddots & \\
& & & \textcolor{blue}{\boldsymbol{D}} \\
\boldsymbol{B}_1 & \boldsymbol{B}_2 & \cdots & \boldsymbol{B}_k
\end{pmatrix}
\begin{pmatrix}
\textcolor{red}{\boldsymbol{D}} & & & \\
& \textcolor{red}{\boldsymbol{D}} & & \\
& & \ddots & \\
& & & \textcolor{red}{\boldsymbol{D}} \\
\boldsymbol{B}_1 & \boldsymbol{B}_2 & \cdots & \boldsymbol{B}_k
\end{pmatrix}
\tag{6}
$$

$$\text{Instance 1} \qquad\qquad \text{Instance 2}$$

**Algorithm 3:** Coefficient refinement algorithm

**Input:** An instance dataset $\{\mathcal{I}_i\}_{i=1}^N$.

1 # Compute the mean and variance
2 **for** $i$ *in* $\{1, \cdots, N\}$ **do**
3     Determine $K$, the number of non-trivial constraints for each block unit. We denote the $k$-th non-trivial constraint in the block unit by $\mathbf{a}_k$
4     **for** $k$ *in* $\{1, \cdots, K\}$ **do**
5         Compute the mean $\mu_k^{(i)}$ and variance $\sigma_k^{(i)}$ across coefficients in $\mathbf{a}_k$ and block units in $\mathcal{I}_i$
6     **end**
7 **end**
8 # When performing mix-up or expansion, given an instance $\mathcal{I}_i$ and a block unit $\mathcal{BU}$
9 **for** $k$ *in* $\{1, \cdots, K\}$ **do**
10     Sample the non-trivial coefficients from $\mathcal{N}(\mu_k^{(i)}, \sigma_k^{(i)})$ to replace those in $\mathcal{BU}$
11 **end**
12 Performing mix-up or expansion

Table 21: The variable, constraint, and edge features used for MILP-StuDio and G2MILP.

| Index | Variable Feature Name | Description |
|---|---|---|
| 0 | Objective | The objective coefficient of the variable |
| 1-4 | Variable type | The variable type, including binary, integer, implicit-integer, and continuous |
| 5 | Specified lower bound | Binary, whether the variable has a lower bound |
| 6 | Specified upper bound | Binary, whether the variable has an upper bound |
| 7 | Lower bound | Binary, whether the variable reaches its lower bound |
| 8 | Upper bound | Binary, whether the variable reaches its upper bound |

| Index | Constraint Feature Name | Description |
|---|---|---|
| 0 | Bias | Normalized right-hand-side term of the constraint |

| Index | Edge Feature Name | Description |
|---|---|---|
| 0 | Coefficient | Constraint coefficient of the edge |

# I More Details on the Data and Experiments

## I.1 Details on Bipartite Graph Representations

The bipartite instance graph representation utilized by MILP-StuDio closely aligns with the approach presented in the G2MILP paper [19]. This representation can be extracted using the observation function provided by the Ecole framework [60]. We list the graph features in Table 21.

## I.2 Details on the Benchmarks

The CA and FA benchmark instances are generated following the process described in [8]. Specifically, the CA instances were generated using the algorithm from [16], and the FA instances were generated using the algorithm presented in [17]. The IP and WA instances are obtained from the NeurIPS ML4CO 2021 competition [32]. The statistical and structural information for all the instances is provided in Table 22.

Table 22: Statistical information of the benchmarks we used in this paper.

|  | CA | FA | IP | WA |
|---|---|---|---|---|
| Constraint Number | 193 | 10201 | 195 | 64306 |
| Variable Number | 500 | 10100 | 1083 | 61000 |
| Block Number | 94 | 100 | 105 | 60 |
| Constraint Type (-Cons) | M, B | M, B | M, B | M, D and DB |
| Variable Type (-Vars) | Bl | Bl | Bl | Bl and Bd |

## I.3   Details on Graph Distributional Similarity

To evaluate the distributional similarity between the training and generated MILP instances, we compute 11 graph statistics [19], as detailed in Table 23. First, we calculate these statistics for both the original training instances and the generated instances. Then, we compute the Jensen-Shannon divergence (JSD) $D_{\text{JS},i}$, for each of the 11 statistics, where $i = 1, \cdots, 11$. The JSD ranges from 0 to $\log 2$, so we standardized the values as follows: $D_{\text{JS},i}^{\text{std}} = \frac{1}{\log 2}(\log 2 - D_{\text{JS},i})$. Finally, we obtain an overall similarity score by taking the mean of the standardized JSD values, score $= \frac{1}{11} \sum_{i=1}^{11} D_{\text{JS},i}^{\text{std}}$. The resulting score falls within the range of $[0, 1]$, where a higher value indicates stronger distributional similarity between the training and generated instances.

Table 23: Statistics for computing structural distributional similarity

| Feature | Description |
|---|---|
| coef_dens | Proportion of non-zero entries in CCM $A$. |
| cons_degree_mean | Average degree of the constraint vertices. |
| cons_degree_std | Standard deviation of the constraint vertex degrees. |
| var_degree_mean | Average degree of variable vertices. |
| var_degree_std | Standard deviation of the variable vertex degrees. |
| lhs_mean | Mean of non-zero entries in CCM $A$. |
| lhs_std | Standard deviation of non-zero entries in CCM $A$. |
| rhs_mean | Mean of the right-hand-side term $\mathbf{b}$. |
| rhs_std | Standard deviation of the right-hand-side term $\mathbf{b}$. |
| clustering_coef | Clustering coefficient of the graph. |
| modularity | Modularity of the graph. |

