# OpenReview forum: "MILP-StuDio: MILP Instance Generation via Block Structure Decomposition"
_NeurIPS.cc/2024/Conference — NeurIPS 2024 poster_

### Official Review · Reviewer_zDfV · 2024-07-08

**Soundness:** 2
**Presentation:** 2
**Contribution:** 2
**Rating:** 5
**Confidence:** 3

**Summary:**

In this paper, the authors note the specific block structures in the constraint coefficient matrices (CCMs) of MILP instances that are closely related to problem formulations, and propose a novel MILP generation framework, called MILP-StuDio. MILP-StuDio identifies blocks in CCMs and decomposes instances into block units that serve as building blocks for MILP instances. New instances are constructed by removing, substituting and appending block units from the original instance. This allows MILP-StuDio to generate approximate instances with high feasibility and computational hardness as existing instances. The authors claim that using instances generated by MILP-StuDio, the learning-based solver is able to significantly reduce the solution time by more than 10%.

**Strengths:**

1. The logical coherence and structural integrity of the manuscript are particularly noteworthy. The organization of the paper with a clear and effective division of content that facilitates understanding and engagement with the material.

2. The code repository and appendices provided by the author are notable for their rich content and contribution to the replicability of the research. The codebase is meticulously organized, enabling a systematic understanding and replication of the study's procedures. The supplementary materials effectively supplement the paper's main content.

3. The paper presents a novel and insightful approach to generating high-quality MILP instances through the exploration of block structures in CCMs. The visualisation of CCMs is particularly compelling, and effectively highlights the central idea of the paper by giving the reader an understanding of how MILP-StuDio's property of preserving block structure works.

4. The robustness of MILP-StuDio is commendable. Even in the absence of clear block structures, as seen in non-structured MILP problems like the set covering problem, the method is able to generate instances that are still usable for downstream tasks. This demonstrates the adaptability and general applicability of the proposed framework.

5. The experimental design is meticulously crafted and effectively demonstrates the advantages of MILP-StuDio. The comparison with both Bowly and G2MILP reveals that MILP-StuDio achieves higher similarity scores while preserving computational hardness and feasibility. Furthermore, the impressive results in improving the performance of learning-based solvers.

**Weaknesses:**

The significantly lower similarity scores for the red and exp operators (Table 7), despite their successful performance in downstream tasks, raises concerns about the appropriateness of the similarity metric for evaluating MILP instance generation. The consistent decline in similarity scores with increasing modification ratios, while maintaining computational hardness and feasibility, further suggests potential limitations of the similarity metric. Could the authors kindly elaborate on their thoughts regarding the role of similarity in MILP instance generation and whether alternative metrics or considerations might be more suitable for this task?

**Questions:**

Please reply to my comment listed in 'Weaknesses'.

**Limitations:**

Not applicable.

---

> ### Author Rebuttal · Authors · 2024-08-07
>
> We thank the reviewer for the positive and insightful comments. We respond to each comment as follows and sincerely hope that our rebuttal could properly address your concerns. If so, we would deeply appreciate it if you could raise your score. If not, please let us know your further concerns, and we will continue actively responding to your comments and improving our submission.
>
> > 1. The lower similarity scores for red/exp and increasing modification ratios suggest limitations of the similarity metric.
>
> Thanks for the valuable insight. The red/exp operators change the instance sizes. The similarity score reflects the statistical similarity between the generated and original instances, considering factors such as density, node degree, coefficients, modularity, etc.
>
> To investigate the similarity score, we first compare the similarity between the original instances with different sizes, which are highly similar in computational hardness and mathematical structures. Using the same setting as in Table 7 in our paper, the results in **Table D1** on the FA dataset show that the similarity scores between instances with different sizes are low. Second, we compare the similarity between the generated instances using exp/red and the original instances with the same sizes (e.g., we compare the similarity between instances generated by red $\eta=0.05$ and original instances with Size 0.95). As shown in **Table D2**, the generated instances achieve high similarity scores.
>
> This suggests that the similarity score has a drawback: **it is sensitive to the graph statistics but fails to capture the information of block structures or problem formulations**, which are crucial determinants of the mathematical properties and computational hardness.
>
> **Table D1**: We compare the similarity between original instances with different sizes. "Size x" refers to instances with x times the size compared to the ones used in the main paper. We use the code provided in [1] to obtain original instances with corresponding sizes.
>
> ||Size 0.95|Size 1.00|Size 1.05|
> |-|-|-|-|
> |**Size 0.95**|1.00|0.34|0.26|
> |**Size 1.00**|-|1.00|0.41|
> |**Size 1.05**|-|-|1.00|
>
> **Table D2**: We compare the similarity between instances generated by red/exp operators and original instances with the same instance sizes.
>
> ||MILP-StuDio (red)|MILP-StuDio (exp)|
> |-|-|-|
> |$\eta=0.05$|0.60|0.62|
>
> > 2. Explain the role of similarity in MILP generation and whether alternative metrics might be more suitable.
>
> It is a good question! To evaluate the generation quality, we may need to incorporate multiple metrics that capture different aspects of the instance properties. Since MILP generation aims to address data inavailability in downstream tasks, we think the **most effective metric is the improvement in downstream tasks brought by the generated instances**.
>
> 1. **The MILP generation field is still in its early stages, and the graph distributional similarity is a simple and intuitive metric to measure the similarity**. The similarity score serves as an important metric in [2], thus we still adopt it as a metric. This situation is analogous to the molecular generation field, where researchers also used metrics like KL divergence and Frechet ChemNet Distance to assess the graph distributional similarity in the early works [3,4].
>
> 2. **We need to incorporate multiple metrics to measure the quality of the instances**. While the graph similarity score cannot reflect the mathematical properties and hardness, computational hardness alone fails to capture the structural information. In molecular generation, researchers compare various metrics, such as validity, uniqueness, and novelty [3,4]. Recently, metrics more related to chemical properties have been proposed, such as atom/molecule stability and drug-likeness [5,6]. **We believe more suitable metrics will be proposed** for MILP generation in the future.
>
>    We propose two metrics more related to the mathematical structures.  (1) **CCM similarity** captures the block structure features, in which we calculate the distance of two digital representations of CCMs. (2) **Constraint distribution** reflects the formulation information. For each instance, we construct a 17-dimensional feature whose components are the proportions of 17 types of constraints proposed in MIPLIB. We then calculate the cosine similarity of the features between two instances.
>
> 3. **The improvement in downstream tasks may be the most effective metric** to test the applicability. Unlike molecular generation, where the goal is to discover novel molecules or drugs, MILP generation aims to **address data unavailability in the downstream tasks**. Therefore, the improvement in downstream task performance **is in line with the initial goal**, while the similarity and computational hardness are intermediate metrics. While existing work [2] mainly focuses on a single task, we try to provide as many tasks as possible to demonstrate the effectiveness comprehensively.
>
>    - Improving the performance of learning-based solvers, including two ML solvers PS (**Table 4 in the paper**) and GNN approach for branching (**Tables 5 and 6**).
>    - Hyperparameter tuning for traditional solvers (**Table 14**).
>    - Hard instances generation for benchmark construction (**Figure 5 in the paper**).
>
>    We show that MILP-StuDio successfully achieves the best performance across all the tested tasks.
>
> [1] Exact Combinatorial Optimization with Graph Convolutional Neural Networks.
>
> [2] A Deep Instance Generative Framework for MILP Solvers Under Limited Data Availability.
>
> [3] Molecule optimization by explainable evolution.
>
> [4] Data-Efficient Molecular Generation with Hierarchical Textual Inversion.
>
> [5] Target Specific De Novo Design of Drug Candidate Molecules with Graph Transformer-based Generative Adversarial Networks.
>
> [6] Geometry-Complete Diffusion for 3D Molecule Generation and Optimization.

---

> > ### Comment · Reviewer_zDfV · 2024-08-10
> >
> > Thanks very much for your reply! I acknowledge that I have read your response.

---

> ### Author Response · Authors · 2024-08-12
>
> Dear Reviewer zDfV,
>
> Thanks for your kind support and for helping us improve the paper. We sincerely hope that our rebuttal has properly addressed your concerns. If so, we would deeply appreciate it if you could raise your scores. If not, please let us know your further concerns, and we will continue actively responding to your comments and improving our work.
>
> Best,
>
> Authors

---

> ### Author Response · Authors · 2024-08-13
> **Extensive studies on the proposed two metrics**
>
> We conducted extensive experiments to study the two metrics we proposed in our rebuttal. We sincerely hope that our rebuttal has properly addressed your concerns. **If so, we would deeply appreciate it if you could raise your scores. If not, please let us know your further concerns, and we will continue actively responding to your comments and improving our work.**
>
> - **CCM Similarity**: To measure the similarity of the Constraint Coefficient Matrices (CCMs) between the original and generated instances, we represented the CCMs using 0-1 matrix representations. The points with value 1 represent the nonzero entries in the CCMs, while the points with value 0 represent the zero entries. We then calculated the matrix similarity between the generated and original instances using the following formula:
>
>   $$\text{CCM similarity}=1-\frac{\\|A_1-A_2\\|_1}{\eta mn}$$
>
>   where $A_1$ and $A_2$ are the 0-1 matrices of the original and generated instances, respectively, $m$ is the number of constraints, $n$ is the number of variables, and $\eta$ is the modification ratio.
>
>   Due to the sparsity of the CCMs, $\eta mn$ can be very large compared to $\\|A_1-A_2\\|_1$. Thus, the CCM similarity is always close to 1, and 0.01 of the CCM similarity can lead to a great difference in the structure. Recall the computational results in Table 3 in our paper, where the performance of MILP-StuDio and G2MILP is the closest in CA, and differ significantly in the other three datasets. This coincides with the results in Table D1, where the CCM similarity between G2MILP and MILP-StuDio is the closest in CA.
>
>   **Table D1**: The average CCM similarity between the generated and original instances.
>
>   |      | G2MILP ($\eta=0.05$) | MILP-StuDio (mix $\eta=0.05$) |
>   | ---- | -------------------- | ----------------------------- |
>   | CA   | 0.95                 | 0.98                          |
>   | FA   | 0.93                 | 0.99                          |
>   | IP   | 0.91                 | 0.99                          |
>   | WA   | 0.87                 | 0.95                          |
>
> - **Constraint distribution.** To capture the differences in the distribution of constraint types, we constructed a 17-dimensional feature vector for each instance, where each component represents the proportion of a specific constraint type proposed in MIPLIB. We then calculated the cosine similarity between the feature vectors of the generated and original instances. Since the modification ratio $\eta$ is small, the cosine similarity is always close to 1, and 0.01 of the cosine similarity can lead to a great difference in the structure. The CCM similarity between G2MILP and MILP-StuDio is the closest in CA, consistent with the results of computational hardness in Table 3 of the paper.
>
>   **Table D2**: The average cosine similarity between the features of the generated and original instances.
>
>   |      | G2MILP ($\eta=0.05$) | MILP-StuDio (mix $\eta=0.05$) |
>   | ---- | -------------------- | ----------------------------- |
>   | CA   | 1.00                 | 1.00                          |
>   | FA   | 99.07                | 1.00                          |
>   | IP   | 99.53                | 1.00                          |
>   | WA   | 99.86                | 1.00                          |

---

> ### Author Response · Authors · 2024-08-14
> **We are looking forward to the reviewers' feedback**
>
> Dear Reviewer zDfV,
>
> Thanks for your kind support and for helping us improve the paper. We are writing to gently remind you that the author-reviewer discussion period will end in less than **12 hours**. We sincerely hope that our response has properly addressed your concerns on the **similarity metric for evaluation**. If so, **we would deeply appreciate it if you could raise your scores**. If not, please let us know your further concerns, and we will continue actively responding to your comments and improving our work.
>
> Best,
>
> Authors

---

### Official Review · Reviewer_Rqw8 · 2024-07-13

**Soundness:** 3
**Presentation:** 2
**Contribution:** 3
**Rating:** 7
**Confidence:** 3

**Summary:**

This paper presents a method for generating instances for mixed
integer programming that mimic the characteristics of existing
instances. The main insight is that the performance characteristics of
MIP solvers relate to the structure of the coefficient matrix of the
instance, with a few examples of such structure given in the paper. So
the new method is designed to discover such structures and replicate
them/combine them in the instances it generates. In experimental
results, the generator seems to produce instances whose behavior is
closer to that of the rest of the family.

**Strengths:**

I liked the overall idea. The paper identifies a feature of instances
that existing generators ignore, proposes a not-too-complicated way to
use it, and the results match the expectations.

Overall a solid paper.

**Weaknesses:**

I think the latter two of the evaluation criteria are
under-explored. Specifically:

- Improving the performance of learning-based solvers: I would expect
  a lot more detail on how these solvers were trained. In general,
  this aspect of the work is not explained very well, I think, and is
  presented as if it should self-evident (which it is not, to me).

- Hard benchmark generation: here there is just a very briefly
  explained generation of a pool of instances that progressively gets
  harder. I do not understand exactly why this test was performed like
  this, what the original instances, nor do I find this a convincing
  argument that the generator can produce instances of increasing
  difficulty.

Finally, I don't like the title of the paper at all. It feels labored
and gimmicky. It is not a reason to reject, though.

various places: operational research -> operations research

110: motivated -> motivating

(3): what is bold O?

**Questions:**

No specific questions

**Limitations:**

Addressed sufficiently

---

> ### Author Rebuttal · Authors · 2024-08-07
>
> We thank the reviewer for the positive and insightful comments. Our rebuttal includes **Tables C1-2 in the attached PDF**. We respond to each comment as follows and sincerely hope that our rebuttal could properly address your concerns. If so, we would deeply appreciate it if you could raise your score. If not, please let us know your further concerns, and we will continue actively responding to your comments and improving our submission.
>
> # W1
>
> >More details on the training of ML solvers
>
> We use **predict-and-search** (PS) [1] and **learning-to-branch** (Branching) [2] ML solvers in this paper. We provide a **brief introduction to the solvers in Appendix D**. We also provide **part of the training and implementation information in Appendix G.1 for PS and Appendix E.1 for Branching**. Here we would like to provide more details to address your concerns.
>
> - **PS**. The implementation and training process align with those in [1]. PS aims to **predict an initial feasible solution for the binary variables**.
>
>   - **Data usage**. The original training and validation set consists of 100 and 20 instances, respectively. We generate 1,000 additional instances to enrich the training set. Then, we run Gurobi on each instance $I$ and collect 50 optimal or near-optimal solutions to approximate the solution distribution $p_i$ for each binary variable of $I$.
>
>   - **Training**. We use a GNN with four half-convolution layers for prediction. To train the GNN predictor $p_\theta(x|I)$, PS adopts the assumption that the variables are independent of each other, $p_\theta(x|I)=\prod_ip_\theta(x_i|I)$. To calculate the prediction target, PS uses the 50 collected solutions of $I$, from which a vector of solution distribution $(p_1,p_2,\dots,p_n)$ is constructed. Here, $p_i=p(x_i=1|I)$ is the probability of variable $x_i$ being assigned the value 1 in $I$. The GNN predictor then outputs the predicted probability $p_\theta(x_i=1|I)$ for each variable. The predictor is trained by minimizing the cross-entropy loss
>
>     $$L(\theta) = -\sum\_{i,j}p_i\log p_\theta(x_i=1|I_j)+(1-p_i)\log (1-p_\theta(x_i=1|I_j))).$$
>
>   - **Hyperparameters**. We set the learning rate to be 0.001, the training epoch to be 10,000, and he batch size to be 8.
>
> - **Branching**. The implementation and training process align with [2]. The experiment settings and results on Branching are in **Appendix E**. Branching is critical in Branch-and-Bound algorithm and can be formulated as a Markov decision process. The branching policy decides to select a variable and partitions its feasible region at each step. **The quality of the selected variables significantly impact the algorithm's efficiency**. The strong branching policy (expert) can select high-quality variables but consumes a long decision time. GNN branching policy serves as a fast approximation.
>
>   - **Data usage**. The training and validation instances are identical to those in PS. Then we run the expert on each instance to collect 11,000 state and selected variable pair $(s,a)$, forming the training dataset $D$.
>   - **Training**. The GNN branching policy $\pi_\theta$ aims to imitate the decision behavior of the expert and is trained by minimizing
>         $$L(\theta)=-\sum_{(s,a)\in D}\log\pi_\theta(a|s).$$
>   - **Hyperparameters**. We set the initial learning rate to be 1e-4, the epoch to be 1,000 with early stopping, and the batch size to be 8.
>
> [1] A GNN-Guided Predict-and-Search Framework for Mixed-Integer Linear Programming.
>
> [2] Exact Combinatorial Optimization with Graph Convolutional Neural Networks.
>
> # W2
>
> >Discussions on the hard instance generation
>
> We would like to give a detailed explanation of the experiment. The test was done in [3], and we followed their settings.
>
> 1. **Hard instance generation** task is **important** as it provides valuable resources to evaluate solvers and thus potentially motivates more efficient algorithms.
> 2. **Experiment settings**. The objective of this experiment is to **test the ability to generate harder instances within a given number of iterations**. We use 30 original instances to construct the pool. In each iteration, for each instance $I$ in the pool, we employ mix/exp on it to generate two new ones, $I'$ and $I''$. We then select the instance among $I,I'$ and $I''$ with the longest solving time to replace $I$ in the pool. This setting is to **preserve the diversity of the pool**. We observe that there exist slight differences in the hardness of the original instances, and the generated instances derived from the harder original instances are also harder than those from easier ones. If we had simply generated 60 instances and selected the hardest 30, the proportion of instances generated from the hard original instances would have continuously increased, reducing the diversity of the pool.
> 3. **More experiment results**. To provide more convincing evidence, we compare with G2MILP in Setcover dataset (G2MILP fails in FA) and find that MILP-StuDio can obtain a harder dataset in the given iterations (**Figure C2**). Additionally, we present the distribution of instance solving times in **Figure C1**, which demonstrates that the solving time becomes progressively longer across the entire distribution during the iterations.
> 4. **Discussions**. The superior performance of MILP-StuDio can be attributed to the strong ability of the mix/exp operators to preserve the hardness. Moreover, exp can generate larger instances and thus are more likely to generate harder ones.
>
> [3] G2MILP: Learning to Generate Mixed-Integer Linear Programming Instances for MILP Solvers.
>
> >The title is labored and gimmicky
>
> Thank you for your valuable suggestion. We will sincerely consider trying to improve the title.
>
> >operational research -> operations research; motivated -> motivating
>
> Thank you very much for reading this article carefully and pointing out the typos. We will fix it in time.
>
> >bold O in (3)
>
> In Eq (3), **O** refers to the matrix with all entries zero.

---

> > ### Comment · Reviewer_Rqw8 · 2024-08-09
> >
> > Thank you for you response. I liked in particular the general comments you gave on the presence of block structure in industrial instances. These comments would make a good addition to the paper.

---

> ### Author Response · Authors · 2024-08-09
>
> Dear Reviewer Rqw8,
>
> Thank you for your kind support and valuable feedback on our paper! We deeply appreciate your insightful comments and constructive suggestions.
>
> Best,
>
> Authors

---

> ### Author Response · Authors · 2024-08-12
>
> Dear Reviewer Rqw8,
>
> Thanks for your kind support and for helping us improve the paper. We sincerely hope that our rebuttal has properly addressed your concerns. If so, we would deeply appreciate it if you could raise your scores. If not, please let us know your further concerns, and we will continue actively responding to your comments and improving our work.
>
> Best,
>
> Authors

---

### Official Review · Reviewer_jbdZ · 2024-07-15

**Soundness:** 2
**Presentation:** 3
**Contribution:** 2
**Rating:** 4
**Confidence:** 5

**Summary:**

This work presents a method for generating MILP instances by leveraging block structure decomposition. The primary aim is to address the challenges of generating high-quality MILP instances that preserve computational properties and structures of the original problems, thereby supporting the study and development of both traditional and learning-based MILP algorithms.

**Strengths:**

Strength:
- The paper is easy to follow
- It is interesting to study instance generation as real-world MIP data is generally limited for the study of learning-based MIP algorithms.
- Although the idea of studying block structure is not new, the focus on block structure decomposition for MILP instance generation makes sense.
- The paper provides numerical experiments on multiple datasets and results looks promising.

**Weaknesses:**

Weakness:
- The paper lacks a detailed discussion on the criteria for selecting subgraphs during block decomposition and manipulation, which could impact the quality of the generated inst
- The proposed method only works on problems with block structures, which limits its broader applicability
- The number of instances on the test bed is very small.
- Code is not provided.

**Questions:**

Question:
- How does the choice of subgraphs during block decomposition and manipulation affect the overall quality and difficulty of the generated MILP instances?
- Could the proposed approach be potentially extended for more general MIP problems that without block structure?

**Limitations:**

Not applicable.

---

> ### Author Rebuttal · Authors · 2024-08-07
>
> Thank you for your valuable comments. We provide **more evidence to show the applicability of MILP-StuDio Global Response**. Our rebuttal includes **Tables A1-6 and B1-5 in the attached PDF**. We sincerely hope that we could properly address your concerns. If so, we would deeply appreciate it if you could raise your score. If not, please let us know your further concerns, and we will continue actively responding to your comments and improving our work.
>
> # W1 & Q1
>
> > How the criteria for selecting subgraphs impact on the quality and difficulty of the generated instances
>
> Thanks for your valuable suggestion. We conduct extensive experiments to investigate the **impact of different policies for selecting subgraphs** during the MILP instance generation process using MILP-StuDio (mix).
>
> The policy we used in our paper is the **random policy**. Given an original instance, we randomly sample a block unit $\mathcal{BU}_{ins}$ (and thus the corresponding subgraph, called instance subgraph) in the instance and randomly sample a block unit $\mathcal{BU}$ (subgraph) from the structure library. We then substitute the instance subgraph with the subgraph sampled from the library.
>
> We explore several subgraph selection policies:
>
> 1. **Random Policy (Rand 1-3)**: To test the robustness of our random policy, we run three different random seeds.
> 2. **Similarity-based Policy (Sim)**: Sim randomly samples an instance subgraph and five block units (subgraphs) from the structure library. Sim then compares the graph similarity score between the instance subgraph and the five subgraphs and then selects the subgraph with the highest similarity to perform the mix-up operation.
> 3. **Reinforcement Learning Policy (RL)**: RL randomly samples an instance subgraph $\mathcal{BU}\_{ins}$ and five block units from the structure library. Each pair of the instance subgraph and subgraph $(\mathcal{BU}\_{ins},\mathcal{BU})$ forms a state, and the five sampled subgraphs are the actions. The reward is designed as the negative absolute value of the difference in computational time between the original and generated instances, $-|t_1-t_2|$.
>
> We compare the performance of these subgraph selection policies in terms of the similarity score, computational hardness, and feasibility ratio of the generated instances in **Table B1**. We also provide the improvement of the downstream learning-based solver PS and GNN branching policy, in **Tables B2 and B3**. Our key observations are:
>
> 1. The performance of the Rand 1-3 policies are close to each other, indicating that our **random policy for subgraph sampling is robust to the random factors**.
> 2. The performance of the Sim and RL policies is slightly better than Rand 1-3, but the difference is not distinct. This implies that **the simple random policy can achieve a satisfactory performance**.
>
> In summary, MILP-StuDio is not overly sensitive to the subgraph sampling policy, and the simple random policy is robust enough to achieve a good performance.
>
> # W2 & Q2
>
> > The proposed method only works on problems with block structures
>
> Thanks for your valuable suggestion.
>
> 1. **MILPs with block structures are commonly seen in real-world applications**, and a large number of works in operations research have studied specific types of MILPs with block structures in the past few decades (**Global Response 1**).
>
> 2. **We believe that our framework can generalize to general MILPs and adjust to broader applicability**. To demonstrate the effectiveness and generalizability of MILP-StuDio, we have also **adjusted our method and conducted experiments on the general non-structural MILP benchmarks in Appendix E.4**. Here we provide more results.
>
>    - To extend our method to general MILP instances, **the framework remains unchanged and we just need to adjust the block decomposition and partition algorithms 1 and 2**. Specifically, the cons&var classification Algorithm 1 is tailored to general MILPs. The graph community reflects the connection of the constraints and variables, which can serve as a generalization of blocks. Block partition and community discovery are both graph node clustering. Thus, we cluster the constraints and variables according to the community and classification results to form generalized 'blocks'.
>
>    - We provide more results on the **Setcover and MIS datasets (two popular benchmarks with non-structural instances)** and further investigate the improvement of the ML solvers. The results demonstrate the outstanding performance of MILP-StuDio on general MILP instances without block structures. (1) **Tables A1 and A3** show that MILP-StuDio can even outperform existing generation methods in terms of the **similarity score and feasibility ratio**. (2) **Tables A1 and A3** show that MILP-StuDio better preserves instances' **computational hardness** with closer solving time to the original instances. (3)  **Tables A2 and A4** show that MILP-StuDio leads to the **greatest improvement for the PS solver**.
>
> # W3
>
> > The number of instances on the test bed is very small.
>
> Thanks for your suggestion. We use a set of 150 testing instances to evaluate the performance of PS and GNN branching policy. The results are reported in **Table B4** for PS and **Table B5** for GNN branching policy. The results on the larger dataset are consistent with those on the small dataset. **MILP-StuDio still leads to the greatest improvement in the performance of the learning-based solvers**.
>
> # W4
>
> We will upload the relevant code to an anonymous link in the official comment.

---

> > ### Comment · Reviewer_jbdZ · 2024-08-10
> >
> > Thanks for the authors' responses. But I still have concerns about applicability of this approach so I will keep my score.

---

> ### Author Response · Authors · 2024-08-12
>
> Dear Reviewer jbdZ,
>
> We sincerely thank you for your valuable comments throughout the review period. We appreciate your time and feedback and would be grateful for any further specifics regarding areas that you believe we have not adequately addressed.
>
> **Regarding the applications of MILPs with block structures.** MILPs with block structures indeed have a wide application in practice. Actually, as we have pointed out in **Appendix C**, the key reasons why MILP instances exhibit block structures can be summarized as follows.
>
> - **Repeated items or entities with similar attributes.** In many real-world applications involving scheduling, planning, and packing problems, we often encounter **multiple items or entities that share the same type or attributes**. For instance, in a scheduling problem, there may be multiple destinations or vehicles that exhibit similar characteristics. Similarly, in a knapsack problem, there can be multiple packages or items that are interchangeable from the perspective of operational research or mathematical modeling.
>
> - **Symmetric interactions between different types of items.** These **repeated items or entities**, as well as their **interactions**, lead to symmetries in the mathematical formulation of the MILP instances. For example, in a scheduling problem, all the vehicles may be able to pick up items from the same set of places and satisfy the demand of the same set of locations.
>
>
>
> **Regarding the applicability of existing methods.** Though wide usage of MILPs with blocks, existing methods fail in these MILPs, leading to infeasibility or degradation of computational hardness. This severely limits the MILP generation method in real-world applications. Our work is a complement of existing methods that are meaningful for the MILP generation field.
>
>
>
> **Regarding the applicability of our method.** We conduct extensive experiments to demonstrate the effectiveness of our method to the general MILP problems with or without block structures. We conduct **six groups of experiments** from **Table A1-6** in the attached pdf, which shows the superiority of MILP-StuDio in terms of the **statistical properties**, mathematical properties, and the **benefits to the downstream tasks**.
>
>
>
> Thanks for your kind support and for helping us improve the paper. We sincerely appreciate your valuable suggestions.

---

> ### Author Response · Authors · 2024-08-13
> **Applicability and Evaluation on the Real-World Dataset**
>
> We would like to express our sincere gratitude once again for your valuable feedback and constructive suggestions. We have made detailed clarifications regarding our applicability. We sincerely hope that our additional response has adequately addressed your concerns. If so, we would greatly appreciate your consideration in raising the score. If there are any remaining concerns, please let us know, and we will continue to actively address your comments and work on improving our submission.
>
> ### Applicability to General MILPs in Multiple Downstream Tasks
>
> - To the best of our knowledge, we are **the first to conduct an extensive study** on the applications of multiple downstream tasks as follows, which gives a comprehensive study of the wide usage of MILP-StuDio. MILP-StuDio achieves **state-of-the-art performance in the three tasks**.
>   - **Improving the performance of learning-based solvers**, including two ML solvers PS (Table 4 in the paper) and GNN approach for branching (Tables 5 and 6 in our paper). This downstream task tests the benefit of a generation method to enhance the performance of a learning-based solver.
>   - **Hyperparameter tuning for traditional solvers** (Table 14 in our paper). This downstream task tests the benefit of a generation method to enhance the performance of a traditional solver. This test is meaningful in industrial applications since the number of high-quality instances used for tuning is a bottleneck for the performance of a traditional solver.
>   - **Hard instances generation for benchmark construction** (Figure 5 in the paper and Figures C1-2 in the attached pdf). Hard instance generation task is important as it provides valuable resources to evaluate solvers and thus potentially motivates more efficient algorithms. We find that MILP-StuDio can obtain a harder dataset in the given iterations than the baselines in Figure C2.
> - MILP-StuDio is **the first method that can be applied to MILPs with block structures**.  Though commonly seen and widely used in real-world applications, MILPs with block structures remain a great challenge for existing generation methods. Existing methods fail to preserve the feasibility and computational hardness of these instances, while MILP-StuDio preserves almost 100% feasibility and comparable computational hardness.
> - MILP-StuDio **can be applied to MILPs without block structures**. We can easily extend our framework to general MILPs as we state in rebuttal.
>
> ### Applicability to MILPs in the Real-world Industrial Dataset
>
> To further demonstrate the effectiveness in real-world applications, we also conduct experiments on a real-world scheduling dataset at an anonymous enterprise, which is **one of the largest global commercial technology enterprises**. The instances **do not** present clear block structures. The results in the following table show that the extended framework generalizes well on the general MILP datasets and has promising potential for real-world applications. The dataset contains 36 training and 12 testing instances. The few training instances reflect the data inavailability problem in real-world applications. We use different data generation methods to enhance the performance of the ML PS solver and list the solving time, objective value, and node number in Table B6 as follows. MILP-StuDio outperforms other baselines in this dataset, highlighting its strong performance and applicability.
>
> | | | Solving time (1000s time limit) |  || Objective |   |   | Node Number |  |
> | -| -| -| - | ------- | - | - | - | - | - |
> |  | PS | G2MILP+PS  | MILP-StuDio+PS | PS | G2MILP+PS | MILP-StuDio+PS | PS   | G2MILP+PS   | MILP-StuDio+PS |
> | instance 1|1509652.00|1509652.00|1509652.00|144.21| 146.67| 131.80 | 38767.00 | 37647.00|39513.00 |
> | instance 2|205197.00| 205197.00|205197.00| 5.83| 9.51| 10.11| 723.00   | 746.00| 1047.00  |
> | instance 3|7483.00 | 7483.00 | 7483.00| 29.50   | 32.68     | 38.49   | 555.00| 751.00| 901.00 |
> | instance 4|447781.00  | 384454.99  | 318675.99 | 1000.00 | 1000.00| 1000.00| 16545.00 | 6201.00 | 5186.00|
> | instance 5|1465601.00 | 1465601.00 | 1465601.00 | 186.85| 81.67| 63.55| 24317.00 | 11481.00| 9732.00 |
> | instance 6| 1293554.00 | 1293554.00  | 1293554.00  | 38.65| 25.41| 32.45| 3471.00  | 2357.00| 3002.00 |
> | instance 7| 1293554.00 | 1293554.00  | 1293554.00 | 38.96| 24.82| 31.76| 3471.00  | 2357.00| 3002.00|
> | instance 8| 612151.00  | 612151.00  | 612151.00| 0.18| 0.35| 0.28| 13.00| 55.00| 13.00 |
> | instance 9| 1578141.00 | 1578141.00 | 1578141.00| 28.53| 26.74|23.61| 3083.00  | 3207.00  | 3150.00  |
> | instance 10| 1149250.00 | 1149250.00|1149250.00| 7.33| 8.89 | 8.05  | 1.00     | 154.00| 1.00  |
> | instance 11| 1030555.00 | 1030555.00 |1030555.00| 0.27|0.35| 0.34 | 1.00| 1.00| 1.00 |
> | instance 12| 1216445.00 | 1216445.00|1216445.00| 23.64| 23.10| 13.83    | 1384.00| 1384.00 | 1.00 |
> | Average     | 984113.66  | 978836.50  | **973354.92**|125.35| 115.02|**112.86**| 7694.25  | 5528.42     | **5462.42**    |

---

> ### Author Response · Authors · 2024-08-13
> **Eagerly await your valuable feedback**
>
> Dear Reviewer jbdZ,
>
> Thanks for your kind support and for helping us improve the paper. We sincerely hope that our rebuttal has properly addressed your concerns. **If so, we would deeply appreciate it if you could raise your scores. If not, please let us know your further concerns, and we will continue actively responding to your comments and improving our work.**
>
> Best,
>
> Authors

---

> ### Author Response · Authors · 2024-08-14
> **Looking forward to your feedback**
>
> Dear Reviewer jbdZ,
>
> We sincerely thank you for your time and efforts during the rebuttal process. We are writing to gently remind you that the author-reviewer discussion period will end in less than **12 hours**. We have responded to your further comments and eagerly await your feedback, and we sincerely hope that our response has properly addressed your concerns. We would deeply appreciate it if you could kindly point out your further concerns about applicability so that we could keep improving our work. We sincerely thank you once more for your insightful comments and kind support.
>
> Best,
>
> Authors

---

### Official Review · Reviewer_jQXE · 2024-07-16

**Soundness:** 2
**Presentation:** 2
**Contribution:** 1
**Rating:** 3
**Confidence:** 4

**Summary:**

The paper presents a novel framework for generating high-quality MILP instances. The proposed method, MILP-StuDio, leverages the block structures in constraint coefficient matrices (CCMs) of MILP instances to preserve computational hardness while allowing scalable and efficient generation. The framework consists of three main steps: block decomposition, structure library construction, and block manipulation through reduction, mix-up, and expansion. Experimental results demonstrate MILP-StuDio's ability to generate instances that improve the performance of learning-based solvers and maintain high similarity to original instances in terms of graph structural distribution and solving properties.

**Strengths:**

The method takes a further look into the block structures to generate MILP instances that share more similarity with the original structures. The empirical results seem better than the previous method G2MILP.

**Weaknesses:**

The proposed MILP-StuDio framework is tailored to highly artificial or synthetic problems with explicit block structures. The paper does not provide a convincing argument or empirical evidence showing that the framework can be generalized in real-world applications. Actually, in most of the real-world applications, most MILP instances do not have such clear block structures at all. MILP instances can have varying and overlapping structures that are not neatly decomposable. The paper is cherry-picking on specific synthetic/artificial datasets (all the four datasets) that have very structured blocks. It’s of no use to generate well-structured data that could be easily manually generated. The paper does not address how the method handles variability in block structures, leading to concerns about its robustness.

**Questions:**

1. Can you provide formal mathematical proofs to support the claim that the MILP-StuDio framework preserves the feasibility and computational hardness of generated instances? How do you theoretically ensure that the generated instances are representative of the original distribution?
2. How does MILP-StuDio handle MILP instances that do not exhibit clear block structures? Can you provide empirical evidence or case studies demonstrating the framework's applicability to a diverse range of MILP problems beyond those with explicit block structures?
3. How do you ensure that the instances generated by MILP-StuDio do not introduce biases that are absent in naturally occurring MILP problems? Can you provide an analysis comparing the distribution and complexity of generated instances with real-world instances?

---

> ### Author Rebuttal · Authors · 2024-08-07
>
> Thank you for your valuable comments. We **show the applicability of MILP-StuDio Global Response**. Our rebuttal includes **Tables A1-6 in the attached PDF**. We sincerely hope that we could properly address your concerns. If so, we would deeply appreciate it if you could raise your score. If not, please let us know your further concerns, and we will continue actively responding to your comments and improving our work.
>
> # W & Q2
>
> >Most MILPs in the real world do not have block structures
>
> **MILPs with block structures have drawn much attention in industrial and academic fields**. They are commonly seen in practice and have been a critical topic in OR (**Global Response 1**). They are challenging for existing generation methods, and MILP-StuDio is **the first method** designed to address them effectively.
>
> >Generalize to general MILPs without clear block structures
>
> **Our method can generalize to MILPs without clear block structures**. In **Appendix E.4**, we conduct experiments on Setcover, a popular benchmark without block structures. We provide further results.
>
> 1. For general MILPs, **we can still use the framework and adjust the block partition Algorithms 2**. Specifically, the cons&var classification Algorithm 1 is tailored to general MILPs. The graph community reflects the connection of the constraints and variables, which can serve as a generalization of blocks. Block partition and community discovery are both graph node clustering. Thus, we cluster the constraints and variables according to the community and classification results to form generalized 'blocks'.
> 2. **We also conduct experiments on MIS without block structures**. We also compare the improvement of downstream ML solvers. The results in **Tables A1-4** show that the extended framework generalizes well on general MILPs and performs best **in instance properties and downstream tasks**.
>
> Thus, MILP-StuDio can be generalized in real-world applications.
>
> # Q1& Q3.2
>
> >Theoretical results for (1) preservation of the feasibility and computational hardness; (2) representativeness of the original distribution
>
> **The theoretical guarantee for these topics remains an open problem**. In operations research, researchers study the theoretical results when specific problem formulations are given. For general MILPs, we do not know the problem structure, and thus theoretical results become extremely hard due to MILPs' complex combinatorial nature.
>
> 1. We try to propose a **feasible condition** for the mix operator. We analyze the BBD structure with the same notations in Sec 4.1. We place the proof in the comment.
>
>    >**Proposition.** Suppose that the original instance P1 is feasible with the form $\min_x\sum_ic_i^Tx_i\, s.t.D_ix_i\le b_i,\sum_iB_ix_i\le b$. We substitute $(c_1,D_1,b_1,B_1)$ with $(c'_1,D'_1,b'_1,B'_1)$ to obtain new ones P2. Suppose further that the matrix $B_1$ and $B'_1$, $D_1$ and $D'_1$ have the same sizes. We define the problem P3 by $\min_x1\,s.t.D'_1x_1\le b'_1,D_1x_1\le b_1,B'_1x_1=B_1x_1.$ **Then the generated instance P2 is feasible if P1 and P3 share a feasible solution of $x_1$**.
>
>    We are sorry that the analysis of computational hardness is much more intractable without knowing the formulations. We are also sorry that we might not understand the definition of representativeness of a distribution in math. We would appreciate it if you could kindly provide more insights into it.
>
>
> >Compare the distribution & complexity between the generated and real instances
>
> 2. Though difficult in theory, we would like to provide empirical results to show the strong performance of MILP-StuDio in the following aspects.
>
>    - **Feasibility and hardness (Complexity)**. Experiments show that MILP-StuDio has the strong ability to preserve feasibility and computational hardness, both in MILPs with and without block structures (**Tables 2 and 3 in the paper; Tables A1 and A3**).
>
>    - **Representativeness (Distribution)**.  We provide graph statistics of the generated and original instances to **compare the distribution and complexity** between them (**Tables A5-6**).
>
>    The results in **Tables A5-6** show that MILP-StuDio can generate instances with similar graph statistics distribution and complexity to the original ones. The high similarity indicates the representativeness of the distribution.
>
> # Q3.1
>
> >Techniques to avoid introducing biases
>
> 1. **Existing work can introduce severe random noise to the structures of CCMs**. In Figures 3 and 7-10, the CCMs of instances generated by existing methods are quite different in matrix structure compared to the original ones.
>
> 2. **We propose a series of techniques to mitigate both structural and coefficient biases**. Different from existing works, MILP-StuDio has the following key features.
>
>    - **We consider the global block structures**. We perform **Algorithms 1 (cons&var classification) and 2 (block partition)** to ensure the high-level block structures (e.g., BD, BBD).
>
>    - **We avoid structural biases introduced by random sampling and inaccurate networks.** Rather than sampling noisy latent variables and using biased ML models for constraint/variable generation, MILP-StuDio leverages collected blocks from the library. The blocks in the library obey the real distribution in the dataset, leading to fewer artificial biases. **Figures 3 and 7-10** show that the generated instances are highly similar to the original ones in terms of CCM structures.
>
>    - **Techniques to avoid coefficient biases**. We find that the distribution of coefficients in the introduced block units may not match that in the original instances (Please see **Appendix G.4**). This may lead to degradation in computational hardness. Thus, we propose the coefficient refinement algorithm **Algorithm 3** to modify the coefficient in block units.

---

> ### Author Response · Authors · 2024-08-12
>
> Dear Reviewer JQXE,
>
> We are writing as the authors of the paper titled "MILP-StuDio: MILP Instance Generation via Block Structure Decomposition" (ID: 14255). Thanks again for your valuable comments and constructive suggestions, which are of great help to improve the quality of our work. As the deadline for the author-reviewer discussion period is approaching (due on Aug 13), we are looking forward to your further comments and/or questions.
>
> We sincerely hope that our rebuttal has properly addressed your concerns. If so, we would deeply appreciate it if you could raise your scores. If not, please let us know your further concerns, and we will continue actively responding to your comments and improving our work.
>
> Best,
>
> Authors

---

> ### Author Response · Authors · 2024-08-13
> **Eagerly await your valuable feedback**
>
> Dear Reviewer jQXE,
>
> We would like to extend our sincere gratitude for the time and effort you have devoted to reviewing our submission. Your positive feedback, insightful comments, and constructive suggestions have been invaluable to us, guiding us in improving the quality of our work!
>
> We are writing to gently remind you that the author-reviewer discussion period will end in less than 36 hours. We eagerly await your feedback to understand if our responses have adequately addressed all your concerns. If so, we would deeply appreciate it if you could raise your score. If not, we are eager to address any additional queries you might have, which will enable us to enhance our work further.
>
> Once again, thank you for your guidance and support.
>
> Best,
>
> Authors

---

> ### Author Response · Authors · 2024-08-13
> **Applicability and Evaluation on the Real-World Dataset**
>
> We would like to express our sincere gratitude once again for your valuable feedback and constructive suggestions. We have made detailed clarifications regarding our applicability. We sincerely hope that our additional response has adequately addressed your concerns. If so, we would greatly appreciate your consideration in raising the score. If there are any remaining concerns, please let us know, and we will continue to actively address your comments and work on improving our submission.
>
>
>
> ### **Applicability to MILPs in the Real-world Industrial Dataset**
>
> To further demonstrate the effectiveness in real-world applications, we also conduct experiments on a real-world scheduling dataset at an anonymous enterprise, which is **one of the largest global commercial technology enterprises**. The instances **do not** present clear block structures. The results in the following table show that the extended framework generalizes well on the general MILP datasets and has promising potential for real-world applications. The dataset contains 36 training and 12 testing instances. The few training instances reflect the data inavailability problem in real-world applications. We use different data generation methods to enhance the performance of the ML PS solver and list the solving time, objective value, and node number in Table B6 as follows. MILP-StuDio outperforms other baselines in this dataset, highlighting its strong performance and applicability.
>
> | | | Solving time (1000s time limit) |  || Objective |   |   | Node Number |  |
> | -| -| -| - | ------- | - | - | - | - | - |
> |  | PS | G2MILP+PS  | MILP-StuDio+PS | PS | G2MILP+PS | MILP-StuDio+PS | PS   | G2MILP+PS   | MILP-StuDio+PS |
> | instance 1|1509652.00|1509652.00|1509652.00|144.21| 146.67| 131.80 | 38767.00 | 37647.00|39513.00 |
> | instance 2|205197.00| 205197.00|205197.00| 5.83| 9.51| 10.11| 723.00   | 746.00| 1047.00  |
> | instance 3|7483.00 | 7483.00 | 7483.00| 29.50   | 32.68     | 38.49   | 555.00| 751.00| 901.00 |
> | instance 4|447781.00  | 384454.99  | 318675.99 | 1000.00 | 1000.00| 1000.00| 16545.00 | 6201.00 | 5186.00|
> | instance 5|1465601.00 | 1465601.00 | 1465601.00 | 186.85| 81.67| 63.55| 24317.00 | 11481.00| 9732.00 |
> | instance 6| 1293554.00 | 1293554.00  | 1293554.00  | 38.65| 25.41| 32.45| 3471.00  | 2357.00| 3002.00 |
> | instance 7| 1293554.00 | 1293554.00  | 1293554.00 | 38.96| 24.82| 31.76| 3471.00  | 2357.00| 3002.00|
> | instance 8| 612151.00  | 612151.00  | 612151.00| 0.18| 0.35| 0.28| 13.00| 55.00| 13.00 |
> | instance 9| 1578141.00 | 1578141.00 | 1578141.00| 28.53| 26.74|23.61| 3083.00  | 3207.00  | 3150.00  |
> | instance 10| 1149250.00 | 1149250.00|1149250.00| 7.33| 8.89 | 8.05  | 1.00     | 154.00| 1.00  |
> | instance 11| 1030555.00 | 1030555.00 |1030555.00| 0.27|0.35| 0.34 | 1.00| 1.00| 1.00 |
> | instance 12| 1216445.00 | 1216445.00|1216445.00| 23.64| 23.10| 13.83    | 1384.00| 1384.00 | 1.00 |
> | Average     | 984113.66  | 978836.50  | **973354.92**|125.35| 115.02|**112.86**| 7694.25  | 5528.42     | **5462.42**    |

---

> ### Author Response · Authors · 2024-08-14
> **Looking forward to your feedback**
>
> Dear Reviewer jQXE,
>
> We sincerely thank you for your time and efforts during the rebuttal process. We are writing to gently remind you that the author-reviewer discussion period will end in less than **12 hours**. We eagerly await your feedback to understand if our responses have adequately addressed your concerns. If so, **we would deeply appreciate it if you could consider raising your score**. If not, please let us know your further concerns, and we will continue actively responding to your comments. We sincerely thank you once more for your insightful comments and kind support.
>
> Best,
>
> Authors

---

### Author Rebuttal · Authors · 2024-08-07

Dear reviewers,

We sincerely thank all reviewers' insightful and constructive comments, which helped to significantly improve our work. We have responded to the comments given by each reviewer in detail. In this global response, we provide a review on the MILPs with block structures and generalization of our framework to general MILPs without clear block structures. We hope this could be helpful for you to better understand our work.

**We explain that the 'Table/Figure x' is the table or figure from the paper, and 'Table/Figure A/B/C x' is in the attached PDF of the author rebuttal.**

# The Importance of MILPs with Block Structures

**The MILPs with block structures are important in industrial and academic fields**. We found that MILP instances with block structures are commonly encountered in practical scenarios and have been an important topic in operations research (OR) with much effort [1-7].

- **MILP with block structures is an important topic in OR**. Analyzing block structures is a critical tool for analyzing the mathematical properties of instances or accelerating the solving process (e.g., Dantzig-Wolfe decomposition [1]) in OR. The MIPLIB dataset also provides visualization results of the constraint coefficient matrices for each instance, highlighting the prevalence of block structures.

- **The MILP instances with block structures are common and have wide applications in daily production and life**. There are many examples where the instances present block structures, including the allocation and scheduling problems  [2], the multi-knapsack problem  [3], the security-constrained unit commitment problem in electric power systems  [4], multicommodity network flow [5], multicommodity transportation problem  [6], vehicle routing problem [7] and so on. In real-world optimization scenarios, there are different types of similar items---such as different workers or machines in planning and scheduling problems, a set of power generation units in the electric power systems, vehicles in the routing problems, and so on---with relevant variables naturally presents a block-structured form in the mathematical models.

- **The datasets we used in this paper (IP and WA) are from real-world applications**. The NeruIPS 2021 Competition of Machine Learning for Combinatorial Optimization released three well-recognized challenging datasets from real-world applications (IP, WA, and the anonymous dataset). Two out of the three competition datasets (IP and WA) have block structures. Moreover, instances from the anonymous dataset are selected from MIPLIB with large parts having block structures. These further reflect the wide application of block structures in real-world applications. Thus, our method indeed works in a wide range of problems in practice.

- **Researchers have investigated specific MILP problems with block structures**. MILP with block structures has a large scope in the optimization field and there has been a wide range of works on specific problems with block structures, and they have developed a suite of optimization problems tailored to these problems. For example, the tailored algorithm for the security-constrained unit commitment problem in electric power systems  [4], multicommodity transportation problem  [6], vehicle routing problem [7], and so on

Thus, MILP with block structures has a large scope in production and optimization. It has drawn much attention in the industry and academic fields.

# Generalize Our Framework to General MILPs

**We believe that our framework can generalize to general MILPs and adjust to broader applicability**. To demonstrate the effectiveness and generalizability of MILP-StuDio, we have also **adjusted our method and conducted experiments on the general non-structural MILP benchmarks in Appendix E.4**. Here we provide more results.

- To extend our method to general MILP instances, **the framework remains unchanged and we just need to adjust the block decomposition and partition Algorithms 1 and 2**. Specifically, the cons&var classification Algorithm 1 is tailored to general MILPs. The graph community reflects the connection of the constraints and variables, which can serve as a generalization of blocks. Block partition and community discovery are both graph node clustering. Thus, we cluster the constraints and variables according to the community and classification results to form generalized 'blocks'.

- We provide more results on the **Setcover and MIS datasets (two popular benchmarks with non-structural instances)** and further investigate the improvement of the ML solvers. The results demonstrate the outstanding performance of MILP-StuDio on general MILP instances without block structures. (1) **Tables A1 and A3** show that MILP-StuDio can even outperform existing generation methods in terms of the **similarity score and feasibility ratio**. (2) **Tables A1 and A3** show that MILP-StuDio better preserves instances' **computational hardness** with closer solving time to the original instances. (3)  **Tables A2 and A4** show that MILP-StuDio leads to the **greatest improvement for the PS solver**.

[1] Decomposition principle for linear programs.

[2] Optimal Allocation of Surgery Blocks to Operating Rooms Under Uncertainty.

[3] Multiple Knapsack Problems.

[4] Security-constrained unit commitment: A decomposition approach embodying Kron reduction.

[5] Multi-Commodity Network Flows.

[6] Multicommodity routing optimization for engineering networks.

[7] The Vehicle Routing Problem.

---

### Decision · Program_Chairs · 2024-09-25

**Decision:**

Accept (poster)

**Comment:**

The work presents an instance generator for mixed-integer linear programs (MILP) with block structure. The reviews were mixed, with some reviewers appreciating the originality of the work, while others raised concerns about its applicability, technical details, and overall impact. The authors' rebuttals were focused on addressing these three key aspects, and they were active and responsive throughout the review process. However, it is unfortunate that only one reviewer (Rqw8) fully engaged in the discussions during and after the rebuttal period.

I have carefully evaluated the paper, the reviews, and the responses. Overall, I am confident that the authors provided comprehensive answers that addressed all the major concerns raised by the review team. Specifically, the authors included experiments on industrial instances, generalized their methodology to a broader class of MILPs, and clarified the technical concerns and general limitations. These changes are also straightforward to incorporate into the main text or to add to the appendix, assuming the authors do implement them.

In conclusion, I believe this is a novel and potentially impactful contribution. Beyond the strong numerical evidence supporting the enhancement of learning-based solvers, it is important to note that block structure is pervasive in mathematical programming and continues to be an area of active research (e.g., in stochastic programming and Benders approaches). The methodology could also be valuable in assessing novel decomposition techniques or in learning general properties associated with real-world instances of interest. The “image representation” is also intriguing and could inspire similar approaches for other structures. Therefore, I do recommend acceptance as I believe this paper is above the bar.